# Cheap2Rich: A Multi-Fidelity Framework for Data Assimilation and System Identification of Multiscale Physics - Rotating Detonation Engines

Yuxuan Bao [* 1]   Jan Zajac [* 2 3]   Megan Powers [4]   Venkat Raman [4]   J. Nathan Kutz [1 2]

## Abstract

Bridging the multi-fidelity gap between computationally inexpensive models and complex physical systems remains a central challenge in machine learning applications to engineering problems, particularly in multi-scale settings where reduced-order models typically capture only dominant dynamics. In this work, we present Cheap2Rich, a multi-scale data assimilation framework that reconstructs high-fidelity state spaces from sparse sensor histories by combining a fast low-fidelity prior with learned, interpretable discrepancy corrections. We demonstrate the performance on rotating detonation engines (RDEs), a challenging class of systems that couple detonation-front propagation with injector-driven unsteadiness, mixing, and stiff chemistry across disparate scales. Our approach successfully reconstructs high-fidelity RDE states from sparse measurements while isolating physically meaningful discrepancy dynamics associated with injector-driven effects. The results highlight a general multi-fidelity framework for data assimilation and system identification in complex multi-scale systems, enabling rapid design exploration and real-time monitoring and control while providing interpretable discrepancy dynamics.

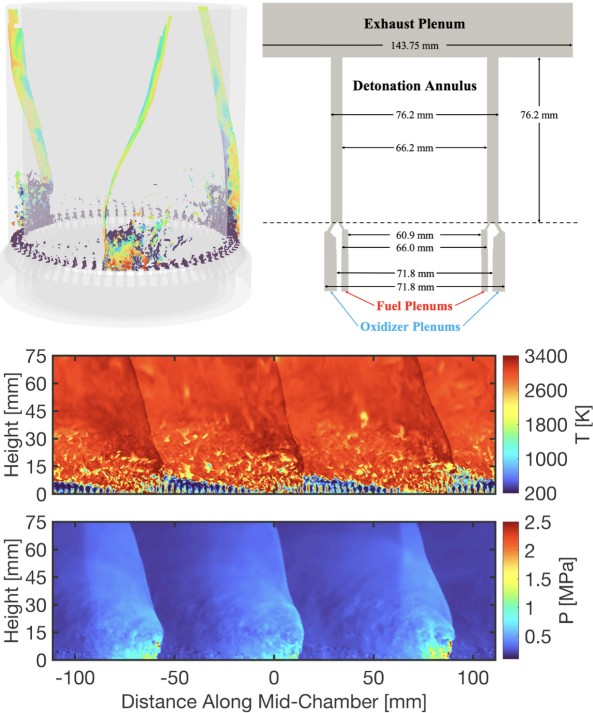

*Figure 1.* A rotating detonation rocket engine (RDRE). Full-scale geometry (top left) and cross-sectional schematic with primary dimensions (top right) Temperature and Pressure mid-channel contour projections (bottom).

## 1. Introduction

Despite significant advancements in physics-informed AI (Kutz et al., 2025; Karniadakis et al., 2021; Brunton

[*]Equal contribution [1]Department of Applied Mathematics, University of Washington, Seattle, USA [2]Department of Electrical and Computer Engineering, University of Washington, Seattle, USA [3]Department of Mathematics, Swiss Federal Institute of Technology Zurich, Zurich, Switzerland [4]University of Michigan, Advanced Propulsion Concepts Lab, Ann Arbor, USA. Correspondence to: J. Nathan Kutz <kutz@uw.edu>.

*Proceedings of the $43^{rd}$ International Conference on Machine Learning*, Seoul, South Korea. PMLR 306, 2026. Copyright 2026 by the author(s).

& Kutz, 2022; Wyder et al., 2025; Fan et al., 2025) and neural operators (Lu et al., 2021; Li et al., 2020; Roy et al., 2025), modeling of multi-scale physics remains a grand challenge due to the inability of methods to model the orders of magnitude different time and space scales present in complex systems. Traditional machine learning methods typically capture the dominant large scale time and space features due to the spectral bias in training (Rahaman et al., 2019). Thus important characteristics and features are effectively band-pass filtered in model training. We introduce a neural network architecture that by construction targets different scales for training. Specifically, Data assimilation (DA) (Ghil & Malanotte-Rizzoli, 1991; Bocquet et al., 2019) provides a principled framework for closing the performance gap in multiscale modeling by combin-

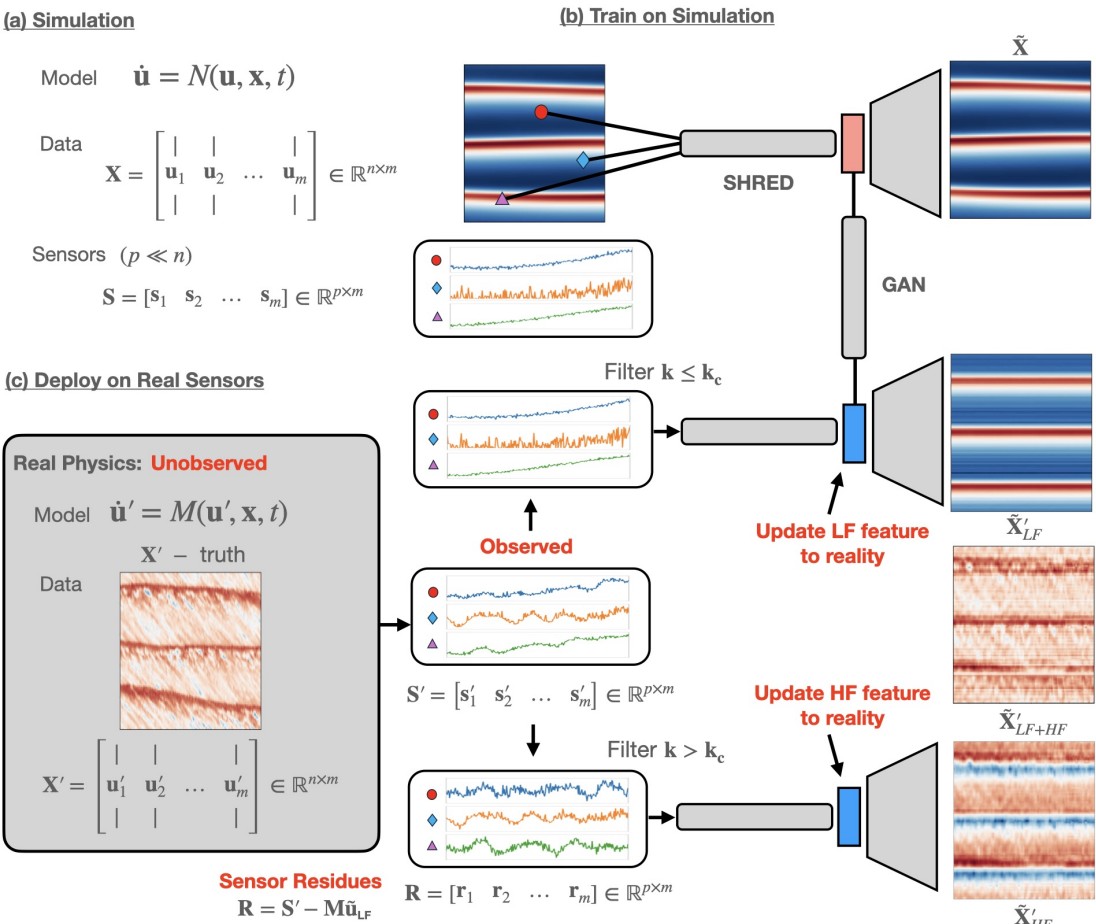

*Figure 2.* The Cheap2Rich architecture. (a) Simulation model provides full-state data $\mathbf{X}$ and sparse sensor measurements $\mathbf{S}$. (b) A standard SHRED network is trained on simulation data to reconstruct full states $\tilde{\mathbf{X}}$ from sensor histories. (c) Deployment on real sensors: the real physics full state $\mathbf{X}'$ is unobserved; only sparse sensor measurements $\mathbf{S}'$ are available. The LF DA-SHRED pathway updates the latent features to reality, producing $\tilde{\mathbf{X}}'_{LF}$. The HF pathway processes sensor residuals $\mathbf{R}$ to capture fine-scale corrections $\tilde{\mathbf{X}}'_{HF}$. The final reconstruction $\tilde{\mathbf{X}}'_{LF+HF}$ combines both pathways to close the multi-fidelity gap.

ing physics-based prediction and experimental observation. Recent advances in latent dynamical system learning (Erichson et al., 2020; Jiang et al., 2025), including the Shallow Recurrent Decoder (SHRED) framework (Williams et al., 2024; Tomasetto et al., 2025; Bao & Kutz, 2025), further enable the discovery of compact latent representations and missing functional structure directly from data. However, the application of DA-enhanced latent learning to strongly nonlinear, multistable systems with sparse, noisy measurements remains largely unexplored (Niu et al., 2024; Liu et al., 2024).

In this work, we introduce a multi-scale data assimilation pipeline that bridges a fast reduced-order RDE model and high-fidelity dynamics using only sparse sensor histories. The method augments a SHRED framework trained on multiscale data with a deployment-time assimilation mechanism that separates the reconstruction into (i) a low-frequency

(LF) component, aligned to the sensor-induced latent distribution via a lightweight model, and (ii) a high-frequency (HF) correction learned directly from sensor-space residuals and regularized to be spectrally sparse. This decomposition enforces a clean division between dominant features that the reduced model already captures and fine-scale discrepancies induced by unmodeled or unresolved physics, yielding a unified surrogate that is both accurate and diagnostically informative.

Rotating detonation engines (RDEs), which are an ideal test case for our method due to the inherent multiscale physics, offer a promising pathway toward high-efficiency propulsion and power generation by sustaining continuous detonation waves within an annular combustor (Sato et al., 2021; 2025; Nakagami et al., 2017; Wolański, 2013). However, the underlying physics of RDEs is governed by tightly coupled compressible flow, shock-detonation interactions, stiff

chemical kinetics, and injector-driven mass, momentum, and energy exchange across disparate spatial and temporal scales (Raman et al., 2023). High-fidelity numerical simulations capable of resolving these coupled processes are computationally prohibitive, often requiring weeks of supercomputing time for a single operating condition (Powers et al., 2026). This computational burden fundamentally limits systematic design exploration, uncertainty quantification, and control-oriented modeling. To address these challenges, reduced-order physics-based models have been developed to capture the dominant detonation-front dynamics at dramatically reduced cost (Koch et al., 2021; Koch & Kutz, 2020; Mendible et al., 2021). Among these, the one-dimensional rotating detonation model of Koch and collaborators has emerged as a widely used surrogate for azimuthal detonation propagation (Koch & Kutz, 2020). While such models accurately represent leading-order detonation physics, they necessarily omit or heavily simplify injector dynamics, mixing processes, time-delay effects, and non-equilibrium losses. As a result, substantial and systematically structured discrepancies persist between low-order model predictions and real experimental RDE data. We show that we can leverage Koch's cheap simulation model (minutes) to approximate simulations of the rich model (weeks), thus allowing for a Cheap2Rich algorithm whereby cheap proxies can be used to model the exceptionally rich multiscale physics observed in reality.

On a three-wave co-rotating configuration (See Fig. 1), the learned LF model remains close to the simple Koch's model prior and cannot reproduce injector-driven variability. Cheap2Rich reduces the error by $74.9\%$ while exhibiting sparse Fourier content concentrated at harmonics of the three-wave structure, indicating phase-locked corrections tied to detonation-front passage and consistent with injector-modulated forcing absent in the baseline model. This accounts for a practical multi-fidelity bridge: the reduced model runs in seconds, and the trained network performs full-state reconstruction from sparse measurements at negligible marginal cost compared to high-fidelity simulation, while the structured HF term provides direct access to the subsequent identification of missing-physics.

Code for this project is available at https://github.com/kro0l1k/Cheap2Rich.

## 2. Preliminaries

### 2.1. RDEs - high fidelity simulation

There has been a lot of prior work on accurate modeling of RDEs that stem from high-fidelity numerical simulations (Raman et al., 2023). In this work, a high-fidelity simulation of the AFRL methane-oxygen rotating detonation rocket engine (RDRE) was studied as shown in Fig.1. This geome-

try was chosen due to the high availability of experimental (Bennewitz et al., 2019) and numerical simulations (Prakash et al., 2021) available. The reacting compressible Navier-Stokes and species transport equations (1) are solved using an in-house compressible flow solver (Sharma et al., 2024) built on the AMReX library (Zhang et al., 2019), which provides the adaptive mesh refinement (AMR) framework.

$$
\begin{aligned}
&\frac{\partial \rho}{\partial t} + \frac{\partial (\rho u_i)}{\partial x_i} = 0, \\
&\frac{\partial (\rho u_i)}{\partial t} + \frac{\partial (\rho u_i u_j)}{\partial x_j} = -\frac{\partial p}{\partial x_i} + \frac{\partial \tau_{ij}}{\partial x_j}, \\
&\frac{\partial (\rho e)}{\partial t} + \frac{\partial (\rho h\, u_j)}{\partial x_j} = \frac{\partial}{\partial x_j}\left(\alpha \frac{\partial T}{\partial x_j}\right) + \frac{\partial (u_i \tau_{ij})}{\partial x_j} \\
&\qquad\qquad + \sum_{k=1}^{N_s} h_k \frac{\partial}{\partial x_j}\left(\rho D \frac{\partial Y_k}{\partial x_j}\right), \\
&\frac{\partial (\rho Y_k)}{\partial t} + \frac{\partial (\rho Y_k u_j)}{\partial x_j} = \frac{\partial}{\partial x_j}\left(\rho D \frac{\partial Y_k}{\partial x_j}\right) + \Omega_k
\end{aligned} \tag{1}
$$

for $k = 1, \ldots, N_s$ where $\rho$ is the density of the fluid, t is time, $x_i$ and $u_i$ are the spatial coordinate and the velocity component in the $i^{th}$ direction, respectively. The viscous stress tensor is given as $\tau_{ij}$ and p is the fluid pressure. The total chemical energy is defined as e, h is the total enthalpy of the mixture, $\alpha$ is the thermal conductivity of the mixture, D is the diffusivity, and T is the temperature. The mass fraction and chemical source term for the $k^{th}$ species are given by $Y_k$ and $\Omega_k$, respectively. These equations are solved in a Cartesian coordinate system using a second-order finite-volume discretization method, and time integration is performed using a strong stability-preserving second-order Runge–Kutta scheme. Detailed finite-rate chemistry is simulated using Cantera with the FFCMy-12 mechanism (Smith et al., 2016; Xu & Wang, 2018) consisting of 12 species and 38 reactions. For additional details on the numerics and implementation, the reader is referred to (Sharma et al., 2024; Carreon et al., 2025; Bielawski et al., 2023).

The high-fidelity simulation employs a nonuniform grid. A fine resolution of 93.6 $\mu$m is used in the injectors, plenums, lower half of the combustion chamber, and in the regions containing the detonation wave. The grid is coarsened to 374 $\mu$m in the upper half of the combustion chamber and further to 748 $\mu$m in the exhaust plenum. The detonation wave is dynamically tagged using the same pressure gradient threshold as (Powers et al., 2026). The resulting cell count for the simulation was 178 million cells and was run using 3500 CPUs. The simulation cost more than 2 million CPU hours and ran for over 2 months. This simulation required 3.2 billion degrees of freedom.

To ignite the RDRE, four high temperature/pressure kernels were equally spaced in the domain and allowed to evolve.

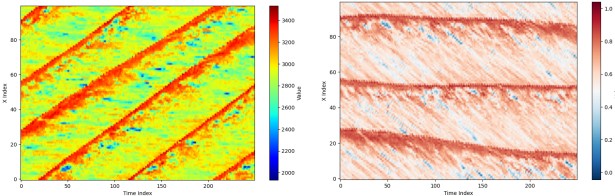

*Figure 3.* Time evolution of 1d projection of temperature contour before preprocessing (left), and in the COM frame of reference after min-max rescaling (right)

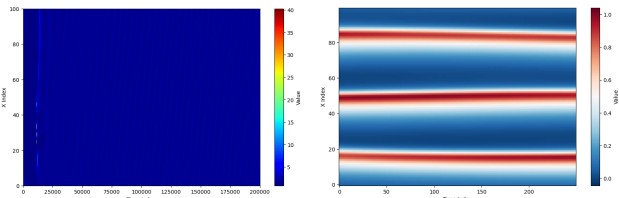

*Figure 4.* Temperature field obtained from Koch's model described in Subsection 2.2 before (left) and after (right) preprocessing.

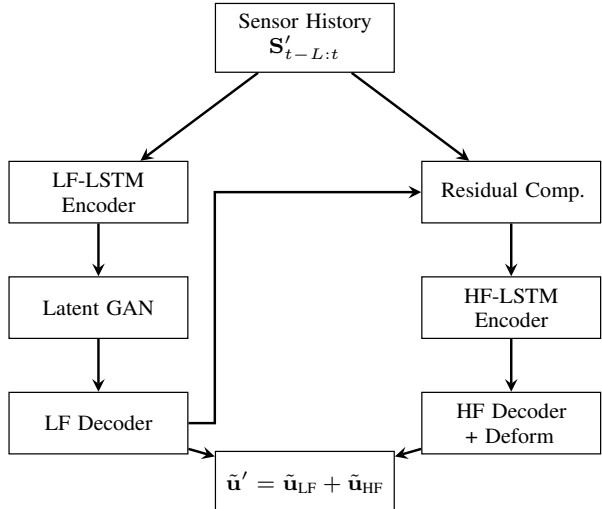

*Figure 5.* Schematic of the Cheap2Rich architecture. The LF pathway learns the dominant dynamics from simulation and aligns to reality via a latent GAN. The HF pathway learns spectrally-sparse corrections from sensor residuals.

After 0.7 ms the simulation reached steady-state and produced three co-rotating waves. The simulation was allowed to continue for additional 0.5 ms of data collection for averaging. Figure 1 shows the unwrapped view that highlights the temperature and pressure contours. Also, the dynamic tagging of the detonation waves are shown in the 3D geometry. 250 snapshots are considered for the following analysis which corresponds to one full rotation of the waves in the detonation annulus. We then project the AMR results first onto a fixed 3d cylindrical mesh with 30000 points and then on a 1d ring with 100 points at the distance from injectors of $20mm$ (See Appendix A).

### 2.2. Koch's One-Dimensional RDE Model

The model described in (Koch & Kutz, 2020) is a 1D model of the reactive Euler equations with source terms representing injection, mixing, and chemical kinetics inside an RDE combustion chamber. We describe the model in detail in the appendix B. With the set of parameters described in 2 it is able to model the three co-rotating waves we observe in the high-fidelity simulation. We select time indices $[100000, 110000]$ which correspond to roughly one rotation of the three waves around the ring, and preprocess them by first subsampling them by taking every 400-th snapshot, leaving exactly 250 timesteps which exactly corresponds to the setting of the high-fidelity simulation. We then cancel out the rotation and scale the temperature field to $[0, 1]$ linearly, obtaining the spatio-temporal field in Fig. 4.

## 3. Cheap2Rich Pipeline

We propose a multi-scale data assimilation framework that decomposes the full-state reconstruction into an additive

low-frequency (LF) backbone and a spectrally-constrained high-frequency (HF) residual. This Cheap2Rich architecture leverages the SHRED and DA-SHRED framework (Williams et al., 2024; Bao & Kutz, 2025) to bridge the multi-fidelity gap by separating the dominant dynamics captured by simplified models from the fine-scale corrections that account for missing physics. The key insight is that enforcing spectral sparsity on the HF correction encourages physically interpretable, parsimonious representations of the discrepancy (Brunton et al., 2016).

### 3.1. Problem Setup

We summarize the notation used throughout this section in Table 3 (Appendix C), following the conventions established in (Bao & Kutz, 2025). The full-state reconstruction is decomposed as

$$\tilde{\mathbf{u}}'(t) = \tilde{\mathbf{u}}_{\text{LF}}(t) + \tilde{\mathbf{u}}_{\text{HF}}(t), \tag{2}$$

where $\tilde{\mathbf{u}}_{\text{LF}}$ captures the dominant low-frequency structure learned from simulation and adapted to reality via latent-space alignment, and $\tilde{\mathbf{u}}_{\text{HF}}$ represents a spectrally-sparse high-frequency correction. The Cheap2Rich architecture consists of two parallel pathways that process the sensor time-history $\{\mathbf{s}'_{t-\ell}\}_{\ell=0}^{L-1}$ with $L$ temporal lags, as illustrated in Figure 5.

### 3.2. Low-Frequency Pathway

The LF pathway follows the standard DA-SHRED methodology (Bao & Kutz, 2025), consisting of a temporal encoder trained on simulation data and a latent-space alignment.

### 3.2.1. TEMPORAL ENCODER

Given sensor history $\mathbf{S}'_{t-L:t} = [\mathbf{s}'_{t-L+1}, \ldots, \mathbf{s}'_t] \in \mathbb{R}^{L \times p}$, the LF encoder maps this sequence to a latent representation with a two-layer LSTM (Hochreiter & Schmidhuber, 1997):

$$\mathbf{z}_{\text{LF}}(t) = \mathcal{E}_{\text{LF}}(\mathbf{S}'_{t-L:t}; \boldsymbol{\theta}_{\text{enc}}) = \text{LayerNorm}\left(\mathbf{h}_L^{(2)}\right), \quad (3)$$

where $\mathbf{h}_L^{(2)} \in \mathbb{R}^{d_z}$ is the final hidden state of the second LSTM layer with hidden dimension $d_z$, and LayerNorm denotes normalization for training stability (Ba et al., 2016).

### 3.2.2. LATENT-SPACE ALIGNMENT VIA GAN

To bridge the distribution shift between simulation-trained latent codes and those induced by real sensor measurements, we employ a residual generator network $\mathcal{G}$ that learns to align the latent distributions:

$$\tilde{\mathbf{z}}_{\text{LF}}(t) = \mathbf{z}_{\text{LF}}(t) + \mathcal{G}(\mathbf{z}_{\text{LF}}(t); \boldsymbol{\theta}_G). \quad (4)$$

The generator $\mathcal{G} : \mathbb{R}^{d_z} \to \mathbb{R}^{d_z}$ is a shallow MLP with LeakyReLU activations, initialized to output near-zero corrections. A discriminator network $\mathcal{D} : \mathbb{R}^{d_z} \to [0, 1]$ is trained adversarially (Goodfellow et al., 2014) to distinguish between latent codes from simulation and transformed codes from real data:

$$\mathcal{L}_D = -\mathbb{E}_{\mathbf{z} \sim p_{\text{real}}} [\log \mathcal{D}(\mathbf{z})] - \mathbb{E}_{\mathbf{z} \sim p_{\text{sim}}} [\log(1 - \mathcal{D}(\tilde{\mathbf{z}}))], \quad (5)$$
$$\mathcal{L}_G = -\mathbb{E}_{\mathbf{z} \sim p_{\text{sim}}} [\log \mathcal{D}(\tilde{\mathbf{z}})], \quad (6)$$

where $\tilde{\mathbf{z}} = \mathbf{z} + \mathcal{G}(\mathbf{z})$ for $\mathbf{z}$ sampled from simulation latents, and the discriminator is trained to classify real sensor latents as real and generator-transformed simulation latents as fake.

### 3.2.3. LF DECODER WITH SPECTRAL CONSTRAINT

The aligned latent code is decoded and then low-pass filtered to enforce the low-frequency constraint:

$$\tilde{\mathbf{u}}_{\text{LF}}(t) = \mathcal{P}_{k_c}\left(\mathcal{D}_{\text{LF}}(\tilde{\mathbf{z}}_{\text{LF}}(t); \boldsymbol{\theta}_{\text{dec}})\right) \in \mathbb{R}^n, \quad (7)$$

where $\mathcal{D}_{\text{LF}}$ is a three-layer MLP with ReLU activations, and $\mathcal{P}_{k_c}$ denotes a low-pass filter that retains only Fourier modes with wavenumber $k \leq k_c$:

$$\mathcal{P}_{k_c}(\mathbf{u}) = \mathcal{F}^{-1}\left(\mathbf{1}_{k \leq k_c} \cdot \mathcal{F}(\mathbf{u})\right), \quad (8)$$

where $\mathcal{F}$ and $\mathcal{F}^{-1}$ denote the discrete Fourier transform and its inverse. This explicit spectral constraint separates LF and HF components (Canuto et al., 2006).

### 3.3. High-Frequency Pathway

The HF pathway is designed to capture the fine-scale discrepancy between the LF reconstruction and reality. The input to the HF pathway is the sensor-space residual between observed measurements and LF predictions at sensor locations. Let $\mathbf{M} \in \mathbb{R}^{p \times n}$ denote the sensor sampling operator. The residual history is computed by subtracting the current LF prediction (at sensor locations) from each lag of the sensor history:

$$\mathbf{r}_t = \mathbf{s}'_t - \mathbf{M}\tilde{\mathbf{u}}_{\text{LF}}(t), \quad t = 0, 1, \ldots, m - 1, \quad (9)$$

yielding the residual time-history $\mathbf{R}_{t-L:t} = [\mathbf{r}_{t-L+1}, \ldots, \mathbf{r}_t] \in \mathbb{R}^{p \times L}$. Note that each lag uses its corresponding LF prediction $\tilde{\mathbf{u}}_{\text{LF}}(t - \ell)$, ensuring the residual captures the discrepancy between observed and predicted states at each time step.

The HF encoder employs an attention mechanism over temporal lags (Bahdanau et al., 2014) to learn which timesteps are most informative for predicting the HF correction, with time-derivative embedding to capture velocity and acceleration information (see Appendix D). The HF decoder generates a base spatial pattern and applies a learned spatially-varying deformation to correct for velocity mismatches.

### 3.4. Training Pipeline

The Cheap2Rich model is trained in four sequential stages to ensure stable learning of each component: (1) SHRED training on simulation data, (2) latent GAN training for distribution alignment, (3) HF-SHRED training with spectral sparsity regularization, and (4) fine-tuning. Complete training details including loss functions and hyperparameters are provided in Appendix E.

### 3.5. Physical Interpretation

The multi-scale decomposition has a natural physical interpretation in the context of RDE dynamics: **LF Component:** Captures the dominant detonation front dynamics that are well-represented by simplified models (e.g., Koch's model). The low-pass filter $\mathcal{P}_{k_c}$ ensures this component contains only large-scale spatial structures with wavenumber $k \leq k_c$. **HF Component:** Represents fine-scale corrections due to unmodeled injector dynamics, mixing processes, and turbulent fluctuations. The spectral sparsity regularization encourages the discovery of parsimonious, dominant correction modes, while the bandlimited penalty discourages energy at very high frequencies. **Time-Delay Embedding:** The temporal derivative augmentation and attention mechanism allow the HF pathway to capture velocity and phase information from the sensor history, enabling correction of wave propagation mismatches between simulation and reality.

## 4. Results

We evaluate the Cheap2Rich framework on the rotating detonation engine dataset described in Section 2. The high-fidelity simulation data, which resolves the full three-dimensional reacting flow including shock dynamics, chem-

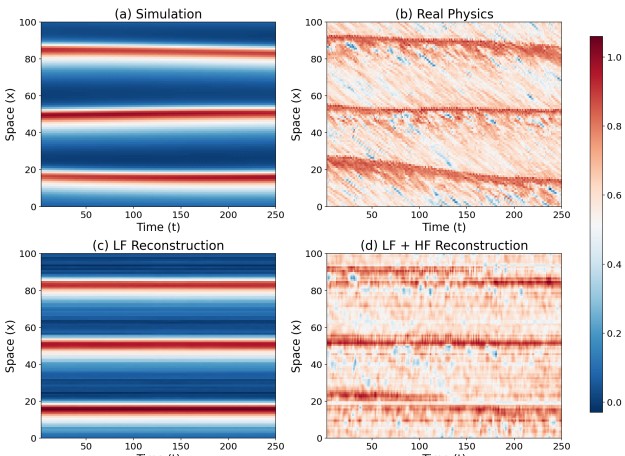

*Figure 6.* Comparison on the full dataset. (a) Koch's model simulation capturing dominant front physics. (b) High-fidelity real physics with complex injector-driven dynamics. (c) LF reconstruction from the GAN-aligned SHRED pathway. (d) Full Cheap2Rich LF+HF reconstruction.

ical kinetics, and injector interactions, requires weeks of supercomputer time to generate for a single operating condition. In contrast, Koch's one-dimensional model captures only the dominant detonation front propagation and can be computed in seconds. Our goal is to demonstrate that Cheap2Rich can bridge this fidelity gap using only sparse sensor measurements from the high-fidelity simulation, without requiring access to its full state.

### 4.1. Experimental Setup

The dataset consists of $m = 250$ temporal snapshots over a spatial domain discretized into $n = 100$ grid points. We use $p = 25$ uniformly distributed sensors and a temporal lag window of $L = 25$ time steps. The data is split into 80% for training and 20% for validation. The cutoff frequency for spectral constraints is set to $k_c = 12$. The training follows a three-stage pipeline: (1) SHRED pretraining on Koch's model simulation, (2) LF-DA-SHRED with latent alignment, and (3) HF training with spectral sparsity ($\lambda = 0.1$, then fine-tuned with $\lambda' = 0.01$).

### 4.2. Reconstruction Performance

Figure 6 presents a qualitative comparison of the reconstruction pipeline. The Koch's model simulation (panel a) captures the three co-rotating detonation fronts but exhibits smooth, idealized dynamics. The high-fidelity simulation (panel b) reveals substantially richer structure: the detonation fronts display spatial variability, and significant fine-scale fluctuations arise from injector dynamics, mixing processes, and turbulent interactions that are absent in the simplified model. The LF reconstruction (panel c) produces output that closely mirrors the Koch's model, as expected

*Table 1.* Reconstruction comparison with neural operator baselines (mean $\pm$ std, 3 runs). All methods receive the same sparse sensor inputs and reconstruct the full spatial field.

| Method | RMSE | SSIM |
|---|---|---|
| Multi-fidelity Gap (no DA) | 0.411 | 0.104 |
| FNO (vanilla) | $0.431 \pm 0.007$ | $0.063 \pm 0.010$ |
| FNO (masked) | $0.367 \pm 0.008$ | $0.258 \pm 0.004$ |
| DeepONet | $0.431 \pm 0.001$ | $0.112 \pm 0.002$ |
| DA-SHRED | $0.414 \pm 0.003$ | $0.134 \pm 0.024$ |
| Cheap2Rich (ours) | $\mathbf{0.103} \pm 0.001$ | $\mathbf{0.364} \pm 0.003$ |

since the LF pathway is trained on simulation data. Despite the GAN-based latent alignment, the LF component alone cannot capture the fine-scale discrepancies, yielding an RMSE of 0.4114. The full Cheap2Rich LF+HF reconstruction (panel d) successfully recovers the complex dynamics of the high-fidelity simulation, reducing the RMSE to 0.1031—an improvement of 74.9%.

### 4.3. Quantitative Analysis

Table 1 summarizes the reconstruction performance across multiple baselines, all operating under identical sparse-sensing constraints: each method is trained on Koch's simulation data and receives only $p = 25$ sensor measurements from the high-fidelity system at inference time, with no access to full high-fidelity states. We evaluate two FNO variants—a vanilla FNO trained on full simulation fields and a masked FNO trained on sparse inputs with an explicit binary mask channel—as well as a DeepONet with a branch network receiving the same time-delay sensor history as our LSTM encoder.

Figure 7 provides the corresponding spatiotemporal reconstructions. FNO (vanilla), which has never seen sparse inputs during training, produces spectrally corrupted outputs (SSIM = 0.063). FNO (masked) achieves SSIM = 0.258 within the simulation distribution but cannot bridge the multi-fidelity gap on high-fidelity sensor inputs. DeepONet reproduces simulation-like spatial structures despite receiving high-fidelity sensor values. DA-SHRED provides minimal improvement over the raw multi-fidelity gap, confirming that the dominant discrepancy lies in unmodeled physics rather than latent distribution mismatch. Cheap2Rich achieves a 74.9% RMSE reduction and 250% SSIM improvement, demonstrating that the multi-scale LF/HF architecture with spectral regularization is essential for bridging the fidelity gap from sparse sensors. Supervised baselines with full high-fidelity state access during training are reported in Appendix F; we note that these operate under a fundamentally different (and easier) information regime.

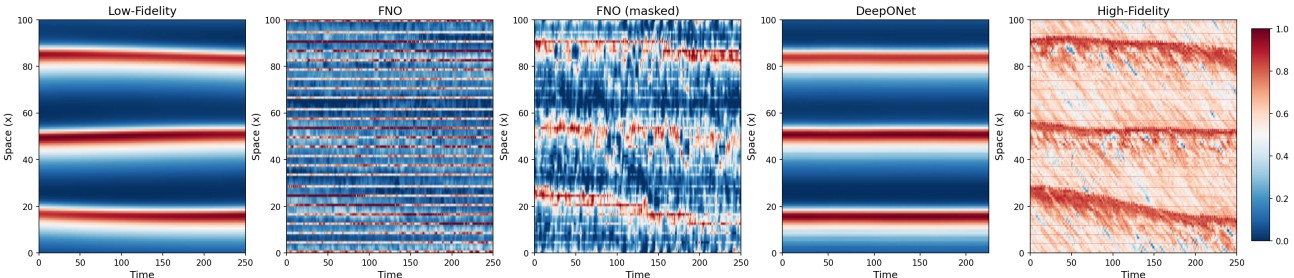

*Figure 7.* Reconstruction comparison of neural operator baselines on the 3-wave co-rotating RDE. From left: Koch's 1D simulation (LF prior), FNO (vanilla), FNO (masked input), DeepONet, and high-fidelity ground truth. FNO and DeepONet fail to bridge the multi-fidelity gap from sparse sensors, reproducing either the smooth simulation manifold or incoherent spatial structure.

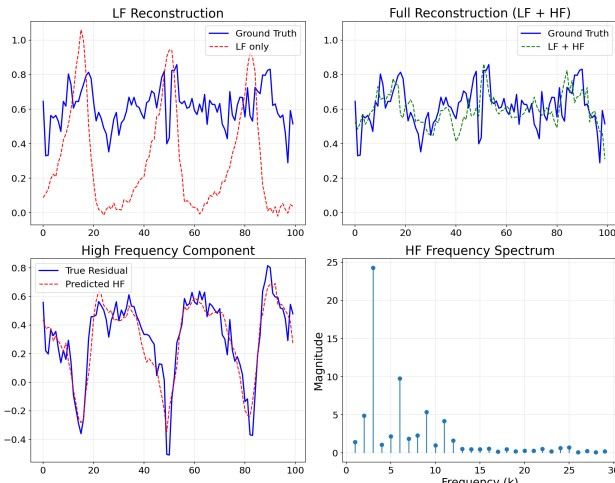

*Figure 8.* Detailed analysis of multi-scale reconstruction. Top-left: LF reconstruction versus ground truth at a representative spatial slice. Top-right: Cheap2Rich LF+HF reconstruction. Bottom-left: Predicted HF component compared to the true residual. Bottom-right: Frequency spectrum of the HF correction showing sparse structure concentrated at wavenumbers $k \in \{3, 6, 9\}$.

### 4.4. Spectral Analysis of the HF Correction

Figure 8 provides detailed analysis of the multi-scale decomposition. The top-left panel shows that the LF reconstruction fails to track the high-fidelity ground truth, while the top-right panel demonstrates that the full Cheap2Rich LF+HF reconstruction closely follows the true dynamics. The bottom-left panel compares the predicted HF component against the true residual (ground truth minus LF prediction), showing good agreement in both amplitude and phase. The bottom-right panel reveals the spectral structure of the learned HF correction. The frequency content is sparse and concentrated at wavenumbers $k \in \{3, 6, 9, 11\}$, with the dominant modes at $k = 3, 6, 9$ corresponding to harmonics of the three co-rotating detonation waves. This physically interpretable structure emerges automatically from the spectral sparsity regularization, suggesting that the HF pathway has learned to represent the injector-driven physics in terms of modes that are commensurate with the underlying waves.

### 4.5. Physical Interpretation

The results demonstrate that the method successfully decomposes the Cheap2Rich gap into interpretable components:

**Front Physics (LF Component).** The LF pathway, trained on Koch's model data and aligned in latents, captures the dominant detonation front locations and propagation speeds. The visible three co-rotating fronts confirm that the LF pathway adequately captures the leading-order dynamics.

**Injector Physics (HF Component).** The HF correction captures the fine-scale dynamics that arise from physics absent in the Koch's model: injector response delays, spatially varying mixing efficiency (Schwer & Kailasanath, 2010), and local turbulent fluctuations (Fotia et al., 2016). These effects manifest as structured corrections concentrated near the detonation fronts and in the inter-front regions where fresh reactants are injected.

The sparse spectral structure of the HF component—with energy concentrated at harmonics of the fundamental three-wave mode—suggests that the injector-driven physics are phase-locked to the detonation front passage. This is physically consistent with the periodic pressure fluctuations that modulate injector flow rates as each detonation wave passes.

### 4.6. Ablation Studies and Robustness

We conduct ablation studies over the spectral cutoff $k_c \in \{3, 6, 9, 12, 15\}$, sensor count $p \in \{5, 10, 15, 25\}$, temporal lag $L \in \{5, 10, 15, 25\}$, and sensor noise level $\sigma \in \{0, 0.02, 0.05, 0.10, 0.15\}$ (Appendix J). Performance improves monotonically with $k_c$ and plateaus at $k_c \geq 12$, confirming robustness to this hyperparameter. The model degrades gracefully under noise, with RMSE increasing from 0.107 to 0.114 at $\sigma = 0.10$, consistent with the spectral cutoff providing noise suppression. RMSE improves steadily with increased spatial coverage, and longer temporal lags yield diminishing returns beyond $L = 15$, consistent with Takens' embedding theory.

To evaluate cross-condition generalization, we train

Cheap2Rich on a two-wave Koch prior ($s = 0.06$) and deploy it on two-wave high-fidelity data with $\sigma = 0.1$ sensor noise (Appendix I), confirming that the multi-fidelity architecture generalizes across different wave-mode configurations and is robust to sensor noise. We further stress-test the framework under a deliberately mismatched prior topology, training on a 3-wave Koch prior and deploying on 2-wave high-fidelity sensors (Appendix I.1). Even in this worst-case scenario, the HF pathway compensates for the topological discrepancy, achieving a 76.9% RMSE reduction. A residual third-wave artifact reduces SSIM relative to the matched case, but the result demonstrates that the HF pathway can absorb substantial structural mismatches. Since Koch's model generates any wave-count configuration in seconds, deploying with a matched prior is trivially inexpensive; the mismatched experiment primarily serves to characterize the limits of the architecture and reveals a diagnostic signal for mode-switch detection.

### 4.7. Computational Implications

The Cheap2Rich framework enables high-fidelity state reconstruction at dramatically reduced computational cost. While the original high-fidelity simulation requires weeks of supercomputer time, our approach requires only sparse sensor measurements from the real system, a low-fidelity baseline model (e.g., Koch's model) that runs in seconds, and lightweight neural network inference that runs in minutes on a standard laptop without requiring GPU clusters. This enables rapid exploration of the RDE design space, where each new operating condition can be characterized through sparse sensing rather than expensive full-scale simulation. The learned HF corrections provide interpretable diagnostics of injector performance without requiring direct measurement of the full injection dynamics.

## 5. Discovery of Missing Physics via SINDy

Beyond closing the simulation-to-reality gap through reconstruction, the Cheap2Rich framework enables explicit identification of the missing physics terms that account for the discrepancy between the simplified Koch's model and high-fidelity dynamics. Following the methodology established in (Bao & Kutz, 2025), we employ sparse identification of nonlinear dynamics (SINDy) (Rudy et al., 2017) within DA-SHRED architecture to discover interpretable governing equations for the learned corrections. The core insight is that the multi-scale decomposition provides natural targets for physics discovery: the LF correction dynamics, the HF component dynamics, and the direct missing physics (see Appendix G for the complete SINDy framework and library construction details).

### 5.1. Discovered Equations

Applying SINDy to the Cheap2Rich outputs yields the following discovered equations.

**Known Baseline: Koch's Model.** For reference, the simplified Koch's model governing the simulation takes the form:

$$\frac{\partial u}{\partial t} + u \frac{\partial u}{\partial x} = q \cdot k(1 - \lambda) \exp\left(\frac{u - u_c}{\alpha}\right) - \epsilon u^2, \quad (10)$$

which captures the detonation front propagation through nonlinear advection and Arrhenius-type reaction kinetics.

**Discovered: LF Correction.** The latent alignment contributes the following correction to the simulation dynamics:

$$\frac{\partial}{\partial t}(u_{\mathrm{LF}} - u_{\mathrm{sim}}) = -0.048\,\Delta u + 0.005\,u_{\mathrm{LF}} - 0.013\,u_{\mathrm{LF}}^2$$
$$-0.056\,u_{\mathrm{LF}} \cdot \Delta u + \mathcal{O}(\partial_x). \quad (11)$$

The dominant terms are polynomial in $u_{\mathrm{LF}}$ and the correction $\Delta u = u_{\mathrm{LF}} - u_{\mathrm{sim}}$, suggesting that the GAN primarily adjusts the amplitude and baseline of the reconstruction rather than its spatial structure.

**Discovered: HF Dynamics.** The high-frequency component evolves according to:

$$\frac{\partial u_{\mathrm{HF}}}{\partial t} = 0.018 - 0.029\,u_{\mathrm{HF}} - 0.018\,u_{\mathrm{LF}} \quad (12)$$
$$-0.067\,u_{\mathrm{LF}} \cdot u_{\mathrm{HF}} + 0.151\,u_{\mathrm{LF}}^2 \cdot u_{\mathrm{HF}} + \mathcal{O}(\partial_x).$$

The coupling terms $u_{\mathrm{LF}} \cdot u_{\mathrm{HF}}$ and $u_{\mathrm{LF}}^2 \cdot u_{\mathrm{HF}}$ indicate that the HF dynamics are modulated by the LF wave structure, consistent with the physical picture of injector-driven fluctuations being phase-locked to detonation front passage.

**Discovered: Direct Missing Physics.** The total discrepancy between real and simulated dynamics (10) is governed by:

$$\frac{\partial}{\partial t}(u_{\mathrm{real}} - u_{\mathrm{sim}}) = 0.723 - 3.79\,u + 6.24\,u^2 - 3.33\,u^3$$
$$+0.403\,u_x - 0.404\,u \cdot u_x + \mathcal{O}(u_{xx}). \quad (13)$$

### 5.2. Physical Interpretation of Discovered Terms

We discovered interpretable corrections to Koch's model, with each term addressing different physical mechanisms.

**Front Physics Corrections.** The direct missing physics equation (13) exhibits Burgers-type structure: a polynomial source term correcting reaction kinetics and advection terms adjusting wave speeds. The cubic polynomial $-3.79u + 6.24u^2 - 3.33u^3$ modifies the effective Arrhenius reaction rates, while $0.40u_x - 0.40u \cdot u_x$ corrects detonation front propagation. These terms primarily address deficiencies in Koch's representation of front physics.

**Injector Physics via LF-HF Coupling.** The HF dynamics equation (12) captures a different mechanism through the

coupling terms $u_{\text{LF}} \cdot u_{\text{HF}}$ and $u_{\text{LF}}^2 \cdot u_{\text{HF}}$. This nonlinear modulation explains the observations from Section 4: HF corrections concentrate near detonation fronts and exhibit spectral sparsity at $k \in \{3, 6, 9\}$. The coupling ensures HF fluctuations are amplified in phase with the three-wave LF structure, consistent with injector-driven physics being modulated by periodic pressure fluctuations.

The discovered equations provide pathways for improving Koch's model without requiring the neural network at inference time. Both direct correction using (13) and hierarchical correction using the LF and HF terms separately are possible; implementation details are provided in Appendix G.

### 5.3. Forward Integration Validation

To confirm that the discovered equations are operable beyond interpretive summaries, we forward-integrate the missing-physics equation (13) on top of Koch's 1D model (Appendix G.4). Using only sparse sensor observations as periodic initial conditions, the six-term polynomial PDE—containing no neural network components—achieves RMSE = 0.139, recovering 68% of the fidelity gap compared to Cheap2Rich's 76%. The rollout remains numerically stable, confirming that the Cheap2Rich decomposition yields actionable model corrections that can be directly embedded into the reduced-order solver without requiring the neural network at inference time.

## 6. Conclusions and Future Works

This work presents a multi-scale data assimilation framework for rotating detonation engines that successfully bridges the multi-fidelity gap between low-fidelity proxy models and high-fidelity complex coupled physics. The proposed Cheap2Rich architecture decomposes the reconstruction task into a low-frequency pathway, which captures dominant detonation front dynamics through latent-aligned representations, and a high-frequency pathway that learns spectrally-sparse corrections from sensor residuals. Applied to a three-wave co-rotating RDE configuration, the framework achieves 74.9% reduction in reconstruction RMSE while requiring only 25 sparse sensor measurements and lightweight neural network inference without GPU clusters. The framework also demonstrates robustness under cross-condition deployment with mismatched prior topology, where the HF pathway compensates for structural discrepancies between the LF prior and the deployment reality. Spectral analysis reveals that the learned HF corrections exhibit sparse structure, with energy concentrated at harmonics of the fundamental three-wave mode, providing physically interpretable representations of injector-driven physics absent from the baseline proxy model. Furthermore, the application of SINDy to the multi-scale decomposition yields explicit governing equations for the missing physics,

distinguishing between front physics corrections of Burgers type and injector physics captured through LF-HF coupling terms. These discovered functionals offer actionable pathways for improving reduced-order simulations.

**Limitations.** We note several limitations of the current work. First, the evaluation is conducted on 1D azimuthal projections of 3D AMR data; extension to full 3D fields requires architectural innovations discussed above. Second, the target "reality" is a high-fidelity simulation rather than experimental sensor data; while the architecture is agnostic to the data source, validation on physical experiments remains future work. Third, while cross-condition generalization has been demonstrated across multi-wave configurations, the high-fidelity dataset for each condition remains constrained by the >2 million CPU-hour simulation cost; larger datasets and additional operating regimes would further strengthen the generality of the framework. Fourth, more extreme topological changes (e.g., co-rotating to pulsing detonation) have not yet been tested. Finally, while the SINDy-discovered equations have been validated via forward integration, incorporating them into a feedback control loop—for example, modulating injector flow rates to maintain stable detonation—remains an important direction for future work.

**Future Directions.** Several promising directions emerge from this work. First, the trained Cheap2Rich model constitutes a computationally efficient surrogate that could enable gradient-based design optimization of RDE geometry, injector placement, and operating conditions at negligible computational cost compared to full-scale simulation. Such surrogate-driven optimization (Forrester et al., 2008) would dramatically accelerate the exploration of high-performance engine configurations that currently require prohibitive computational resources. Second, Cheap2Rich's ability for full-state reconstruction from sparse real-time sensor measurements suggests potential applications in closed-loop control of RDE systems. By coupling the surrogate model with reinforcement learning algorithms (Rabault et al., 2019), one could develop controllers that modulate injector mass flow rates (Fotia et al., 2016) to guarantee stable detonation with a prescribed number of waves (Bennewitz et al., 2018), addressing a critical challenge in transitioning RDE technology from laboratory demonstrations to operational propulsion systems. Third, while the present work demonstrates Cheap2Rich on 1D RDE dynamics, the LF/HF decomposition with spectral regularization is not domain-specific; concurrent work has extended the paradigm to 2D remote sensing applications, suggesting broader applicability to turbulent flows, climate monitoring, and other multi-scale systems where multi-fidelity approaches could reduce the computational burden of high-resolution simulation.

## Impact Statement

In this paper we investigate the applications of Machine Learning to the frontier of propulsion engineering - rotating detonation engines. The current bottlenecks in the design and engineering of RDEs stem from prohibitive simulation costs and our work demonstrates progress in finding inexpensive surrogate models which allow for design and control of these systems. We acknowledge that RDE technology is dual-use, with potential applications in both civilian and defense propulsion systems. The efficiency gains from rotating detonation engines and the reduced computational barriers enabled by surrogate models such as Cheap2Rich could accelerate the development of cleaner, more efficient aerospace propulsion and power generation systems.

## Acknowledgments

LLMs from OpenAI and Anthropic have been used in preparation of the code and corrections to the manuscript. This work was supported in part by the US National Science Foundation (NSF) AI Institute for Dynamical Systems (dynamicsai.org), grant 2112085. JNK further acknowledges support from the Air Force Office of Scientific Research (FA9550-24-1-0141). The authors also acknowledge from the AFOSR/AFRL Center of Excellence in Assimilation of Flow Features in Compressible Reacting Flows under award number FA9550-25-1-0011, monitored by Dr. Chiping Li and Dr. Ramakanth Munipalli. MP acknowledges support from the National Defense Science and Engineering Graduate (NDSEG) Fellowship, USA through the Air Force Research Laboratory (AFRL).

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

## A. Preprocessing of the High Fidelity Sim

### A.1. Interpolation of ADR onto a fixed spatial grid

To map each snapshot from the native (unstructured) simulation point cloud onto a fixed cylindrical reference grid with $3 \times 100 \times 100$ points equally spaced in the $r, \phi, z$ directions in a cylindrical coordinate system, we build a KD-tree over the source coordinates $\{\mathbf{x}_j\}_{j=1}^{N_s} \subset \mathbb{R}^3$ and, for every target grid point $\mathbf{g}_i$ ($i = 1, \ldots, N_g$), query its $k$ nearest neighbors $\mathcal{N}_k(i)$ with Euclidean distances $d_{ij} = \|\mathbf{g}_i - \mathbf{x}_j\|_2$. The field value $v(\mathbf{g}_i)$ is then obtained by inverse-distance weighting with a softened exponent, i.e.,

$$\hat{v}(\mathbf{g}_i) = \sum_{j \in \mathcal{N}_k(i)} w_{ij}\, v(\mathbf{x}_j), \quad w_{ij} = \frac{(d_{ij} + \varepsilon)^{-1/2}}{\sum_{\ell \in \mathcal{N}_k(i)} (d_{i\ell} + \varepsilon)^{-1/2}},$$

where $\varepsilon$ is a small constant for numerical stability; this yields a smooth local interpolation while remaining computationally efficient via batched neighbor queries.

### A.2. Projection onto a 1d ring

Finally, we pick $h_* = 20\,\mathrm{mm}$ and take the mean of the three points of the grid at the fixed $h_*$ and $\phi(i) = \frac{i}{100}2\pi$ obtaining a 1d dataset which is presented on Figure 3. After the projection step, we also counter the rotation of the waves by shifting the frame at time index $i$ by $i\frac{2\pi}{250}$ radians, and changing the Temperature scale with a min-max scaler.

## B. Koch's Model

The physical system describes a compressible reactive flow where fuel and oxidizer are continuously injected into an annular combustor channel, mix at a rate governed by a mixing parameter $s$, and undergo chemical reaction according to Arrhenius kinetics with Damköhler number $Da$ and activation energy $Ea$. The system conserves mass, momentum, energy, and a progress variable (mixture fraction $z$) that tracks the degree of mixing and reaction. Key physical phenomena include injection through the boundary characterized by an area ratio $AR$, heat release $hv$ from chemical reactions controlled by an ignition temperature $T_{\mathrm{ign}}$, and a choked flow condition at the injection boundary that depends on the local pressure state. The governing equations employ a gamma-law equation of state with specific heat ratio $\gamma = 1.29$, representative of detonation products, and include Heaviside functions to model activation of injection, chemical reactions, and boundary conditions.

The numerical solution employs a finite volume method with operator splitting to separately handle hyperbolic transport and stiff reactive source terms. The inviscid flux terms are discretized using a second-order Clawpack framework (Ketcheson et al., 2012) with an HLLC (Harten-Lax-van Leer-Contact) Riemann solver that resolves shock waves, contact discontinuities, and expansion fans in the reactive flow field. The source terms arising from injection, mixing, and chemical reaction are integrated using a second-order explicit Runge-Kutta method (RK2) with adaptive time stepping controlled by a CFL condition of 0.1. The chemical source term incorporates a temperature-dependent reaction rate with Arrhenius kinetics, activated only above a threshold temperature of 1.01×Tign to ensure numerical stability. Periodic boundary conditions are enforced at both domain boundaries to simulate the azimuthal periodicity of the annular combustor, and the simulation tracks primitive variables (density, velocity, pressure, temperature, and mixture fraction) over a domain length of 24 characteristic lengths discretized with 100 grid points, evolving the solution to a final dimensionless time of 180 to capture multiple detonation wave passages and establish quasi-steady periodic behavior.

Four distinct RDE configurations were investigated to demonstrate the universality of the model 9. Just by changing the $s$ parameter in the model we can obtain simulations which have quasi-steady states of one ($s = 0.05$) or two waves ($s = 0.06$), a pulsing detonation ($s = 0.04$), and three waves which correspond to the high-fidelity setting with ($s = 0.07$).

We find a set of parameters which best resemble the data observed in the high resolution 3d simulation. For the following set of parameters we are able to achieve a quasi-steady state of three co-rotating waves 2.

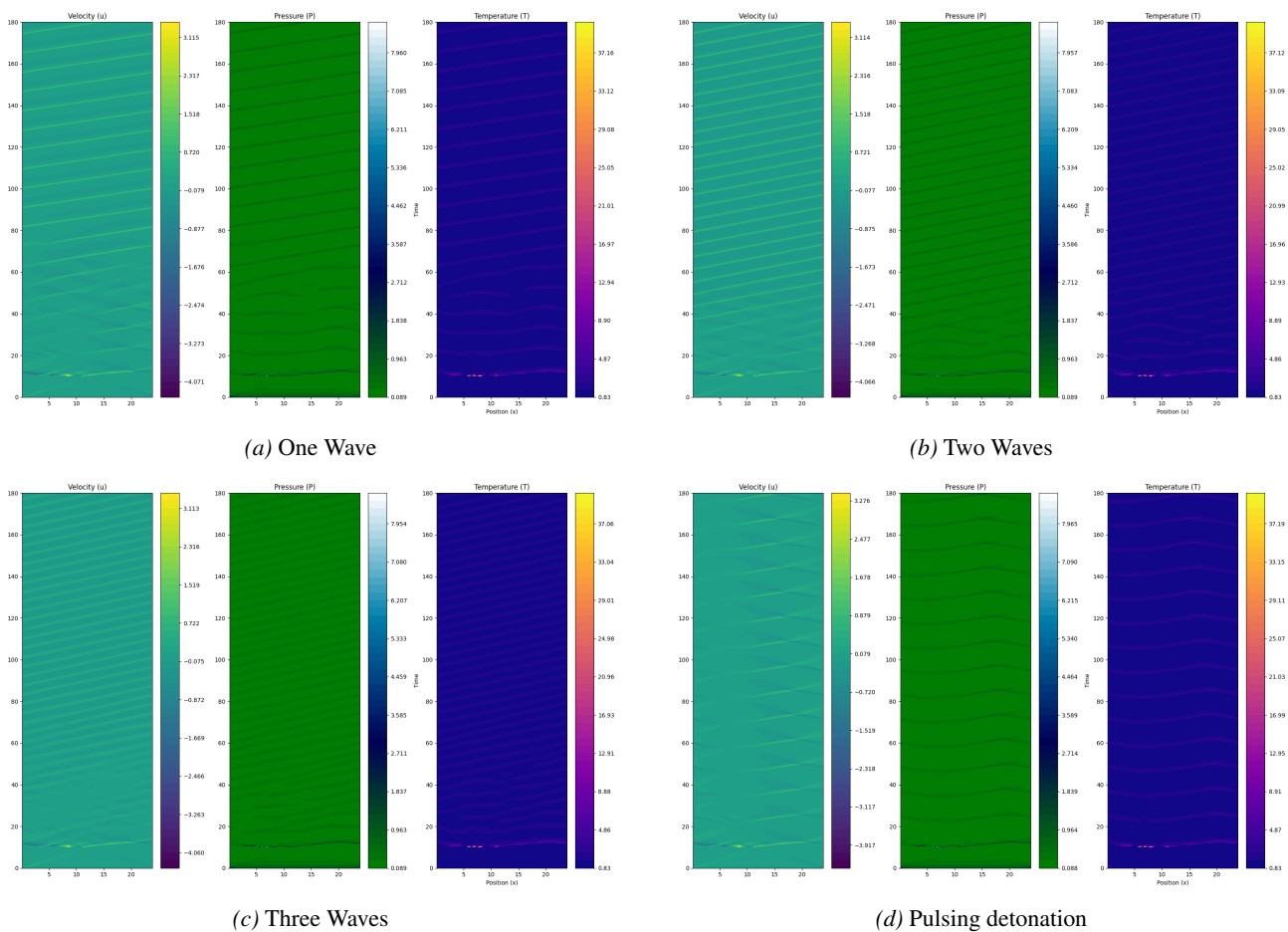

*(a)* One Wave

*(b)* Two Waves

*(c)* Three Waves

*(d)* Pulsing detonation

*Figure 9.* Comparison of four RDE operational modes demonstrating model capacity to simulate different wave structures

## C. Notation Summary

See Table 3.

## D. High-Frequency Pathway Details

This appendix provides detailed specifications of the high-frequency pathway components described in Section 3.

### D.1. Time-Derivative Embedding

To capture velocity and acceleration information critical for correcting phase mismatches, the residual history is augmented with finite-difference temporal derivatives:

$$\dot{\mathbf{r}}_\ell = \frac{\mathbf{r}_{\ell+1} - \mathbf{r}_{\ell-1}}{2\Delta t}, \tag{14}$$

$$\ddot{\mathbf{r}}_\ell = \frac{\mathbf{r}_{\ell+1} - 2\mathbf{r}_\ell + \mathbf{r}_{\ell-1}}{\Delta t^2}, \tag{15}$$

forming an augmented input $\tilde{\mathbf{R}} = [\mathbf{r}_\ell, \dot{\mathbf{r}}_\ell, \ddot{\mathbf{r}}_\ell] \in \mathbb{R}^{L \times 3p}$ that embeds the local temporal structure.

| Parameter | Value | Parameter | Value |
|-----------|-------|-----------|-------|
| $\gamma$ | 1.29 | $Da$ | 289 |
| $p_{\text{ref}}$ | 1.0 | $T_{\text{ign}}$ | 5.8 |
| $\rho_{\text{ref}}$ | 1.0 | $h_v$ | 24.6 |
| $R$ | 1.0 | $E_a$ | 11.5 |
| $T_{\text{ref}}$ | 1.0 | $L$ | 24.0 |
| $AR$ | 0.2 | $m_x$ | 100 |
| $s$ | 0.07 | $t_{\text{final}}$ | 180.0 |
| $\beta$ | 14.286 | | |

*Table 2.* Simulation parameters for three co-rotating waves configuration.

|  | Simulation | Reality |
|--|-----------|---------|
| State Space | $\mathbf{u}_k = \mathbf{u}(\mathbf{x}, t_k) \in \mathbb{R}^n$ | $\mathbf{u}'_k = \mathbf{u}'(\mathbf{x}, t_k) \in \mathbb{R}^n$ |
| Sensor Data | $\mathbf{s}_k = \mathbf{M}\mathbf{u}_k \in \mathbb{R}^p$ | $\mathbf{s}'_k \in \mathbb{R}^p$ |
| Data Matrix | $\mathbf{X} = [\mathbf{u}_1 \cdots \mathbf{u}_m] \in \mathbb{R}^{n \times m}$ | $\mathbf{X}' = [\mathbf{u}'_1 \cdots \mathbf{u}'_m] \in \mathbb{R}^{n \times m}$ |
| Sensor Matrix | $\mathbf{S} = [\mathbf{s}_1 \cdots \mathbf{s}_m] \in \mathbb{R}^{p \times m}$ | $\mathbf{S}' = [\mathbf{s}'_1 \cdots \mathbf{s}'_m] \in \mathbb{R}^{p \times m}$ |
| Sampling Operator | $\mathbf{M} \in \mathbb{R}^{p \times n}$ | |
| Governing Equations | $\mathbf{u}_t = \mathcal{N}(\mathbf{u}, \mathbf{x}, t)$ | $\mathbf{u}'_t = \mathcal{M}(\mathbf{u}', \mathbf{x}, t)$ |

*Table 3.* Summary of variables, data, and models used in the Cheap2Rich formulation. The state space is of dimension $n$, there are $m$ snapshots of temporal measurements using $p$ sensors. For reality, only sensor measurements $\mathbf{s}'_k$ are observed; the full state $\mathbf{u}'_k$ is never directly available.

## D.2. Temporal Attention Encoder

The HF encoder employs an attention mechanism over temporal lags to learn which timesteps are most informative for predicting the HF correction:

$$\mathbf{H} = \text{LSTM}_{\text{all}}(\tilde{\mathbf{R}}; \boldsymbol{\theta}_{\text{all}}) \in \mathbb{R}^{L \times d_z}, \tag{16}$$

$$\mathbf{h}_{\text{final}} = \text{LayerNorm}\left(\text{LSTM}_{\text{main}}(\tilde{\mathbf{R}}; \boldsymbol{\theta}_{\text{main}})\right) \in \mathbb{R}^{d_z}, \tag{17}$$

$$\boldsymbol{\alpha} = \text{softmax}\left(\mathcal{A}(\mathbf{h}_{\text{final}}; \boldsymbol{\theta}_A)\right) \in \mathbb{R}^L, \tag{18}$$

$$\mathbf{z}_{\text{HF}} = \sum_{\ell=1}^{L} \alpha_\ell \mathbf{H}_\ell, \tag{19}$$

where $\text{LSTM}_{\text{all}}$ returns hidden states at all timesteps, $\text{LSTM}_{\text{main}}$ is a two-layer LSTM returning only the final hidden state, $\mathcal{A} : \mathbb{R}^{d_z} \to \mathbb{R}^L$ is a two-layer attention network with Tanh activation, and $\boldsymbol{\alpha}$ represents learned attention weights over temporal lags.

## D.3. HF Decoder with Spatial Deformation

The HF decoder generates a base spatial pattern and applies a learned spatially-varying deformation to correct for velocity mismatches:

$$\mathbf{u}_{\text{base}}(t) = \gamma \cdot \mathcal{D}_{\text{HF}}(\mathbf{z}_{\text{HF}}(t); \boldsymbol{\theta}_{\text{HF}}) \in \mathbb{R}^n, \tag{20}$$

where $\gamma$ is a learnable scale parameter initialized to $0.5$.

To handle spatially and temporally varying wave velocities, we apply a deformation-based correction:

$$\boldsymbol{\delta}(t) = \tanh\left(\mathcal{F}_{\text{shift}}(\mathbf{z}_{\text{HF}}; \boldsymbol{\theta}_{\text{shift}})\right) \cdot \delta_{\max} \in \mathbb{R}^n, \tag{21}$$

$$\mathbf{a}(t) = \frac{1}{2}\text{Softplus}\left(\mathcal{F}_{\text{amp}}(\mathbf{z}_{\text{HF}}; \boldsymbol{\theta}_{\text{amp}})\right) + \frac{1}{2} \in \mathbb{R}^n, \tag{22}$$

where $\boldsymbol{\delta}$ is a position-dependent shift field bounded by $\delta_{\max}$ grid points (typically $\delta_{\max} = 10$), and $\mathbf{a}$ is a positive amplitude modulation field centered around 1.

The final HF output is obtained via spatial warping with periodic boundary conditions:

$$u_{\text{HF}}(x_i) = a_i \cdot u_{\text{base}} \left( (x_i + \delta_i) \mod L_x \right), \tag{23}$$

where bilinear interpolation is used for non-integer shifts to ensure smoothness.

## E. Training Pipeline Details

This appendix provides complete details of the four-stage training pipeline for the Cheap2Rich model.

### E.1. Stage 1: SHRED Training on Simulation

A standard SHRED model is first trained on simulation data with sparse sensor inputs. The model consists of an LSTM that maps sensor histories to a latent space, followed by a decoder MLP:

$$\mathcal{L}_{\text{Stage 1}} = \frac{1}{m} \sum_{k=1}^{m} \|\mathcal{D}_{\text{LF}} \left( \mathcal{E}_{\text{LF}}(\mathbf{S}_{t_k - L:t_k}) \right) - \mathbf{u}_k\|_2^2, \tag{24}$$

where $\mathcal{E}_{\text{LF}}$ denotes the LSTM encoding and $\mathcal{D}_{\text{LF}}$ denotes the decoder. This pretrained SHRED model captures the dominant dynamics of the inexpensive simulation and serves as the LF pathway in the multi-scale architecture.

### E.2. Stage 2: Latent GAN Training

The generator $\mathcal{G}$ and discriminator $\mathcal{D}$ are trained to align latent distributions:

$$\min_{\boldsymbol{\theta}_G} \max_{\boldsymbol{\theta}_D} \ \mathcal{L}_D + \mathcal{L}_G, \tag{25}$$

where latent codes are extracted from both simulation and real sensor data using the frozen LF encoder.

### E.3. Stage 3: HF-SHRED Training with Spectral Sparsity

The HF pathway is trained with sensor-only supervision combined with a spectral sparsity regularizer. Let $k_c$ denote a user-specified cutoff frequency. The training objective is:

$$\mathcal{L}_{\text{Stage 3}} = \mathcal{L}_{\text{sensor}} + \lambda \mathcal{R}_{\text{freq}}(\tilde{\mathbf{u}}_{\text{HF}}) + \mu \mathcal{L}_{\text{mag}}, \tag{26}$$

where each term is defined as follows.

**Sensor Residual Loss.** The HF output must match the observed residual at sensor locations:

$$\mathcal{L}_{\text{sensor}} = \|\mathbf{M}\tilde{\mathbf{u}}_{\text{HF}}(t) - \mathbf{r}_t\|_2^2. \tag{27}$$

**Bandlimited Spectral Sparsity.** Let $\hat{u}_{\text{HF}}(k)$ denote the discrete Fourier coefficients of $\tilde{\mathbf{u}}_{\text{HF}}$ obtained via rFFT. The regularizer encourages sparsity within a specified frequency band while penalizing energy in higher frequencies:

$$\mathcal{R}_{\text{freq}}(\tilde{\mathbf{u}}_{\text{HF}}) = \underbrace{\frac{\sum_{k=0}^{k_c} |\hat{u}_{\text{HF}}(k)|}{\sqrt{\sum_{k=0}^{k_c} |\hat{u}_{\text{HF}}(k)|^2 + \epsilon}}}_{\text{sparsity within } k \leq k_c} + \beta \underbrace{\frac{\sum_{k>k_c} |\hat{u}_{\text{HF}}(k)|^2}{\sum_{k \geq 0} |\hat{u}_{\text{HF}}(k)|^2 + \epsilon}}_{\text{high-frequency energy ratio}}, \tag{28}$$

where $k_c$ is a user-specified cutoff frequency and $\beta \gg 1$ (typically $\beta = 100$) penalizes energy above this cutoff to encourage spectrally sparse corrections within the $[0, k_c]$ band while suppressing energy above $k_c$. The cutoff $k_c$ defines a shared spectral search region rather than a hard LF/HF partition: the LF output is spectrally smooth due to the spectral bias of neural networks (Rahaman et al., 2019), and the HF pathway discovers sparse corrections the smooth LF missed within this same band. The out-of-band penalty prevents the HF pathway from absorbing sensor noise, as confirmed by the noise robustness study in Appendix J.

For the RDE experiments with $p = 25$ sensors over a domain of $n = 100$ grid points, we set the cutoff frequency to $k_c = 12$, which empirically balances reconstruction fidelity with spectral parsimony.

**Magnitude Constraint.** To prevent the HF component from dominating:

$$\mathcal{L}_{\text{mag}} = \left[\max\left(0, \|\tilde{\mathbf{u}}_{\text{HF}}\|_1 - \tau\right)\right]^2, \tag{29}$$

where $\tau$ is estimated from the sensor residual scale.

**Warmup Schedule.** The sparsity weight $\lambda$ is ramped from $0$ to its target value over an initial warmup period to allow the model to first learn the basic residual structure before enforcing sparsity.

### E.4. Stage 4: Fine-Tuning

The HF pathway is fine-tuned with reduced sparsity weight $\lambda' = 0.1\lambda$ to allow greater expressivity while maintaining the learned sparse structure.

### E.5. Inference and Full-State Reconstruction

At inference time, given a new sensor history $\mathbf{S}'_{t-L:t}$, the full-state estimate is computed as:

$$\mathbf{z}_{\text{LF}} = \mathcal{E}_{\text{LF}}(\mathbf{S}'_{t-L:t}), \tag{30}$$
$$\tilde{\mathbf{z}}_{\text{LF}} = \mathbf{z}_{\text{LF}} + \mathcal{G}(\mathbf{z}_{\text{LF}}), \tag{31}$$
$$\tilde{\mathbf{u}}_{\text{LF}} = \mathcal{P}_{k_c}\left(\mathcal{D}_{\text{LF}}(\tilde{\mathbf{z}}_{\text{LF}})\right), \tag{32}$$
$$\mathbf{r}_\ell = \mathbf{s}'_{t-\ell} - \mathbf{M}\tilde{\mathbf{u}}_{\text{LF}}, \quad \ell = 0, \dots, L-1, \tag{33}$$
$$\tilde{\mathbf{u}}_{\text{HF}} = \text{Deform}\left(\mathcal{D}_{\text{HF}}(\mathcal{E}_{\text{HF}}(\mathbf{R}_{t-L:t}))\right), \tag{34}$$
$$\tilde{\mathbf{u}}'(t) = \tilde{\mathbf{u}}_{\text{LF}}(t) + \tilde{\mathbf{u}}_{\text{HF}}(t). \tag{35}$$

### E.6. Hyperparameter Configuration

Table 4 summarizes the default hyperparameters.

### E.7. Computational Cost and Model Complexity

Table 5 summarizes the computational cost and model complexity.

*Table 5.* Computational cost and model complexity.

| Metric | Value |
|---|---|
| *Model Complexity* | |
| LF-SHRED parameters | 4.9 K |
| HF-SHRED parameters | 163.9 K |
| Total parameters | 168.8 K |
| *Training Time* | |
| Stage 1 (LF-SHRED) | 7.92 sec |
| Stage 2 (GAN) | 1.09 sec |
| Stage 3 (HF-SHRED + Fine-tuning) | 38.87 sec |
| Total | 47.88 sec |
| *Hardware* | |
| CPU | Apple M4 Pro |
| Memory | 24 GB |

*Table 4.* Default hyperparameters for the Sparse-Frequency DA-SHRED architecture.

| Component | Parameter | Value |
|---|---|---|
| Data | Number of sensors | 25 |
| | Temporal lags $L$ | 25 |
| | Train/Valid split | 80%/20% |
| LF-SHRED Encoder | Hidden dimension $d_z$ | 32 |
| | LSTM layers | 2 |
| | Dropout rate | 0.1 |
| | Normalization | LayerNorm |
| LF Decoder | Hidden layers | [128, 128] |
| | Activation | ReLU |
| GAN | Generator hidden | 64 |
| | Discriminator hidden | 64 |
| | Activation | LeakyReLU |
| HF-SHRED | Hidden dimension | 32 |
| | LSTM layers | 2 |
| | Dropout rate | 0.1 |
| | Lag attention | True |
| | Attention hidden | 64 |
| | Velocity correction | Deformation |
| | Max spatial shift | $\pm 10$ grid points |
| HF Decoder | Hidden layers | [128, 128] |
| | Deformation net | [128, $N$] |
| | Amplitude net | [64, $N$] |
| | Scale $\gamma$ init | 0.5 |
| Sparsity | $\lambda_{\text{sparse}}$ | 0.1 |
| | Out-of-band penalty $\beta$ | 100.0 |
| | Fine-tune $\lambda'_{\text{sparse}}$ | 0.01 |
| Optimization | Batch size | 32 |
| | Optimizer | Adam / AdamW |

# F. Comparison to other baselines

This appendix benchmarks our approach against standard supervised baselines. We assume access to paired low-fidelity and high-fidelity datasets, and consider the reconstruction task

$$\mathbb{R}^p \ni s_t \ \mapsto \ u'_t \in \mathbb{R}^n,$$

where $s_t$ denotes the $p$ sensor measurements at time $t$ and $u'_t$ the corresponding high-fidelity state. We compare (i) SHRED models that use a history of sensor measurements (lagged inputs) and (ii) feed-forward MLPs that map sensors to state instantaneously. For each architecture, we evaluate two training strategies: training solely on high-fidelity data, and pretraining on low-fidelity data followed by fine-tuning on high-fidelity data.

All models are evaluated with an RMSE calculated on the final 50 snapshots of the high-fidelity dataset, held out from training, as well as SSIM evaluated at the entire high-fidelity dataset. When pretraining is used, we train on 250 preprocessed snapshots from Koch's model before fine-tuning on the high-fidelity data.

**Important distinction.** The supervised baselines in this appendix section have access to full high-fidelity states in their training loss—a fundamentally different and easier problem setting than Cheap2Rich, which operates without any full-state supervision from the high-fidelity system. These baselines are included for reference to contextualize reconstruction quality;

direct comparison of RMSE values across problem settings is not appropriate. On structural fidelity (SSIM), Cheap2Rich (0.364) still substantially outperforms all supervised baselines (best: MLP at 0.213).

### F.1. SHRED trained on high-fidelity data only

We use $p = 25$ sensors, $L = 5$ lags, and a latent dimension $d_z = 32$. The model is trained for 500 epochs with Adam (initial learning rate $10^{-3}$), achieving RMSE $= 0.102237$ on the test set and SSIM of $0.171135$ on train + test. To enable inference for $t \leq L$, we pad the sensor histories in both the training and test sets with zeros at the initial time steps. Reconstructions are shown in Figure 10.

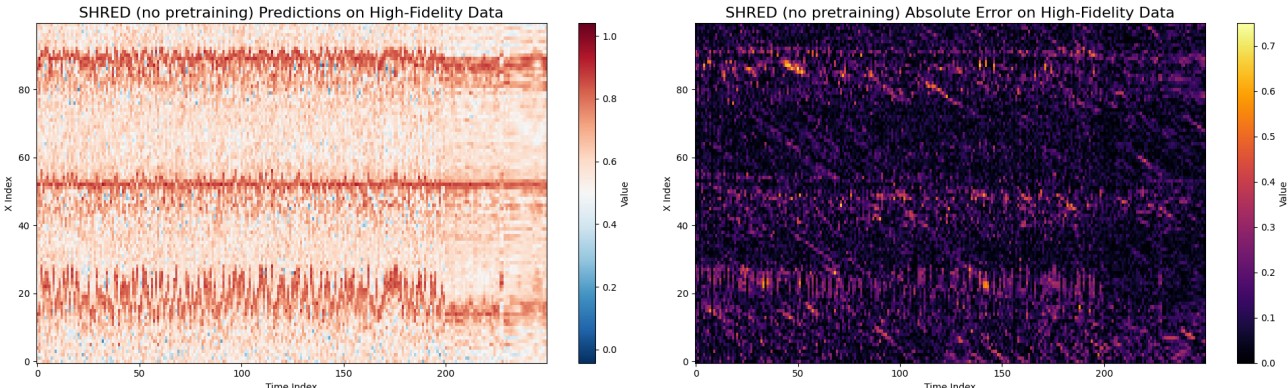

*Figure 10.* SHRED trained on high-fidelity data only.

### F.2. SHRED pretrained on Koch's model, then fine-tuned on high-fidelity data

We again use $p = 25$, $L = 5$, and $d_z = 32$. The model is pretrained for 500 epochs on Koch's data and then fine-tuned for 300 epochs on the high-fidelity data using Adam (initial learning rate $10^{-3}$). This yields RMSE $= 0.107945$ on the test set, and SSIM of $0.176757$ on train + test. As above, sensor histories are zero-padded to support inference for $t \leq L$. Reconstructions are shown in Figure 11.

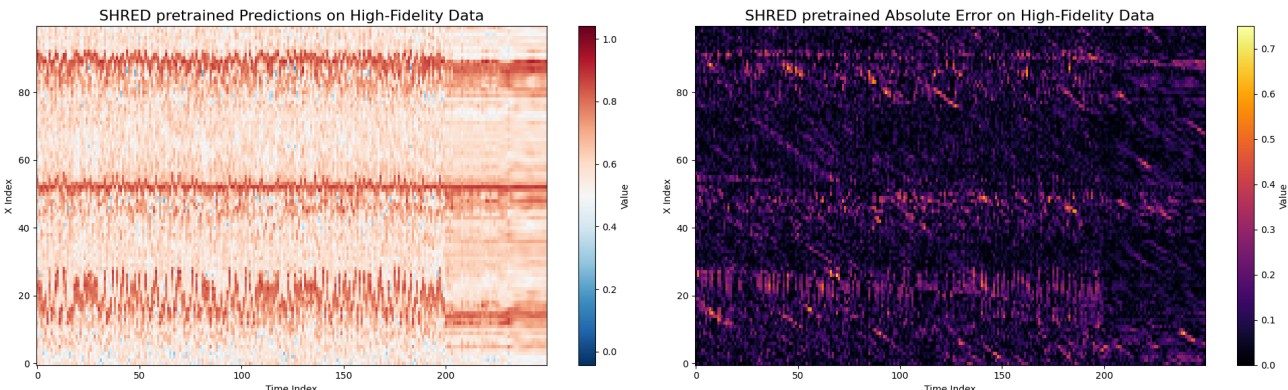

*Figure 11.* SHRED pretrained on Koch's data and fine-tuned on high-fidelity data.

### F.3. MLP trained on high-fidelity data only

We use $p = 25$ sensors and a feed-forward MLP with layer widths $[25, 128, 128, 128, 100]$ and ReLU activations. The model achieves RMSE $= 0.093735$ on the test set and SSIM of $0.213414$ on train + test. Reconstructions are shown in Figure 12.

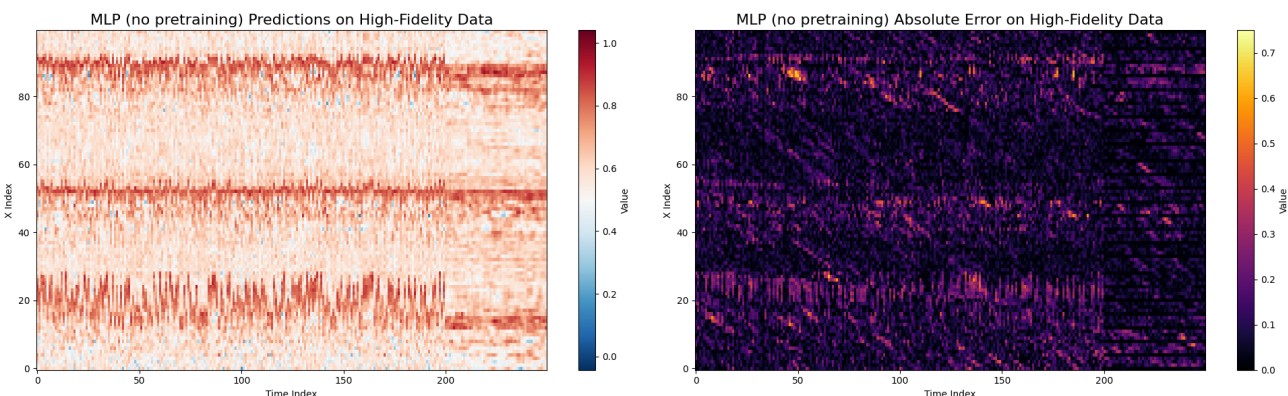

*Figure 12.* MLP trained on high-fidelity data only.

### F.4. MLP pretrained on Koch's model, then fine-tuned on high-fidelity data

We use the same MLP architecture as above. The model is pretrained for 500 epochs on Koch's data and then fine-tuned for 300 epochs on the high-fidelity data with Adam (initial learning rate $10^{-3}$), achieving RMSE $= 0.107945$ on the test set, and SSIM of $0.213414$ on train + test. Reconstructions are shown in Figure 13.

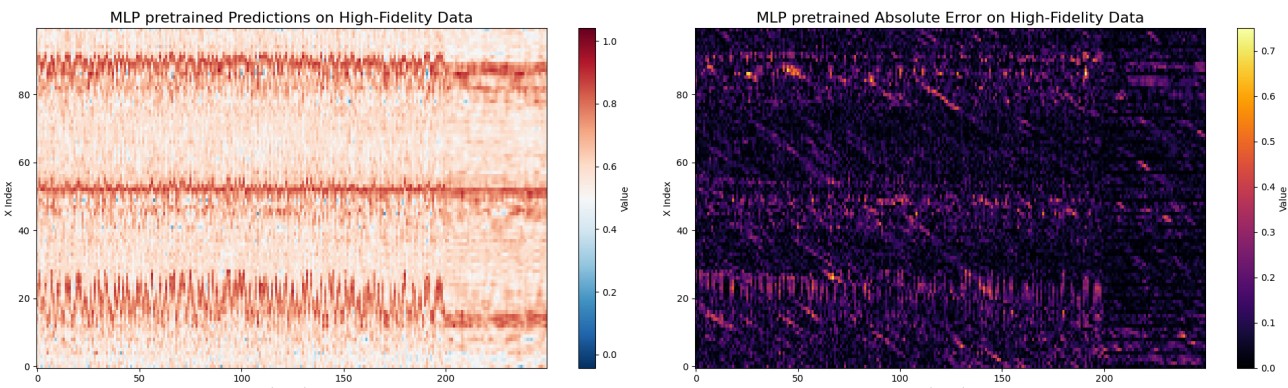

*Figure 13.* MLP pretrained on Koch's data and fine-tuned on high-fidelity data.

### F.5. Cheap2Rich (ours)

Figure 14 reports reconstructions produced by Cheap2Rich. The test-set error is RMSE $= 0.1031$, and the SSIM on the entire high-fidelity dataset is $0.3638$. While the aggregate RMSE is comparable to the baselines, the reconstructions better preserve the salient spatiotemporal structure, which is also explained by the much higher SSIM. In particular, the three dominant wavefronts are sharper and more consistently separated, and fine-scale features are recovered more faithfully, including the colder regions near $(x, t) = (80, 25)$ and $(45, 75)$.

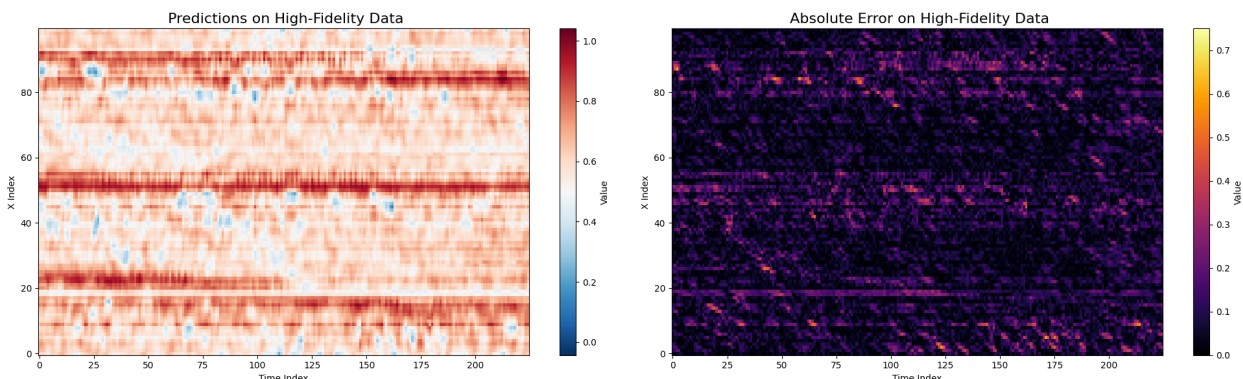

*Figure 14.* Cheap2Rich reconstructions on the high-fidelity test set.

# G. SINDy Framework and Simulation Integration Details

This appendix provides additional details on the SINDy-based physics discovery described in Section 5.

### G.1. SINDy Framework for Discrepancy Modeling

Given the LF reconstruction $\tilde{\mathbf{u}}_{\mathrm{LF}}$, the HF correction $\tilde{\mathbf{u}}_{\mathrm{HF}}$, and the known Koch's simulation $\mathbf{u}_{\mathrm{sim}}$, we can pose three complementary discovery problems:

**LF Correction Dynamics.** The GAN-based latent alignment induces a correction to the simulation output. We seek to identify the functional form of this correction:

$$\frac{\partial}{\partial t}\left(\tilde{\mathbf{u}}_{\mathrm{LF}} - \mathbf{u}_{\mathrm{sim}}\right) = f\left(\Delta u, \Delta u_x, u_{\mathrm{LF}}, u_{\mathrm{LF},x}, \ldots\right), \tag{36}$$

where $\Delta u = \tilde{\mathbf{u}}_{\mathrm{LF}} - \mathbf{u}_{\mathrm{sim}}$ denotes the LF correction field.

**HF Component Dynamics.** The high-frequency pathway captures fine-scale physics absent from the Koch's model. We identify its governing dynamics:

$$\frac{\partial \tilde{\mathbf{u}}_{\mathrm{HF}}}{\partial t} = g\left(u_{\mathrm{HF}}, u_{\mathrm{HF},x}, u_{\mathrm{LF}}, u_{\mathrm{LF}} \cdot u_{\mathrm{HF}}, \ldots\right). \tag{37}$$

**Direct Missing Physics.** Most directly, we can identify the total discrepancy between real and simulated dynamics:

$$\frac{\partial}{\partial t}\left(\mathbf{u}_{\mathrm{real}} - \mathbf{u}_{\mathrm{sim}}\right) = h\left(u, u_x, u_{xx}, u \cdot u_x, \ldots\right). \tag{38}$$

where $u$ denotes $\mathbf{u}_{\mathrm{sim}}$.

### G.2. Library Construction and Sparse Regression

For each discovery problem, we construct a library of candidate nonlinear terms $\boldsymbol{\Theta}$ and solve the sparse regression problem:

$$\frac{\partial \mathbf{u}}{\partial t} = \boldsymbol{\Theta}(\mathbf{u}, \mathbf{u}_x, \mathbf{u}_{xx}, \ldots)\boldsymbol{\xi}, \tag{39}$$

where $\boldsymbol{\xi}$ is a sparse coefficient vector recovered via sequential thresholded least squares (STLSQ) (Brunton et al., 2016).

The candidate library is tailored to the physical context of RDE dynamics. For the direct missing physics discovery, we use:

$$\boldsymbol{\Theta} = \left[1,\ u,\ u^2,\ u^3,\ u_x,\ u_{xx},\ u \cdot u_x,\ \Delta u,\ \Delta u_x,\ \ldots\right], \tag{40}$$

where spatial derivatives are computed spectrally via FFT to ensure accuracy (Brunton & Kutz, 2022). The library is normalized column-wise before regression to ensure fair coefficient comparison across terms of different magnitudes.

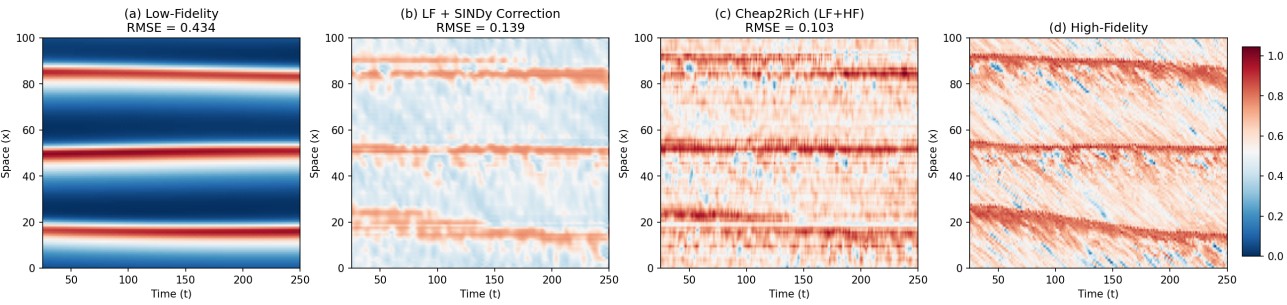

*(a)* Spatiotemporal reconstruction. The SINDy-corrected Koch model (b) recovers the dominant correction structure (RMSE = 0.139, 68% improvement over LF), while Cheap2Rich (c) additionally captures fine-scale amplitudes and structures (RMSE = 0.103, 76%).

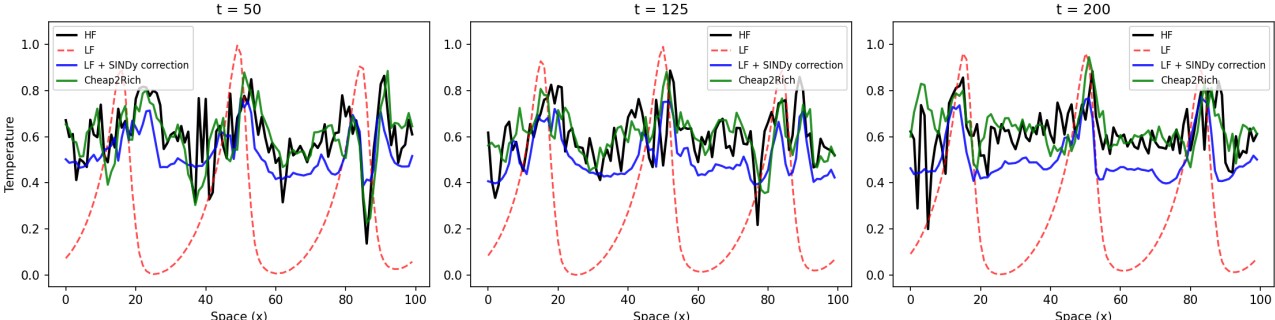

*(b)* Spatial slices at representative time steps. The SINDy correction shifts the LF solution toward the correct amplitude range; Cheap2Rich further recovers the sharp, localized features.

*Figure 15.* Forward integration of the SINDy-discovered missing-physics equation (Eq. 13) on top of Koch's 1D model. The six-term polynomial PDE contains no neural network components yet recovers 68% of the fidelity gap, validating the system identification claim.

The STLSQ algorithm iteratively performs least-squares regression and thresholds small coefficients to zero, promoting sparsity while maintaining accuracy. We employ adaptive thresholding, starting from $\alpha = 0.001$ and incrementing until the discovered equation contains 2–8 active terms, balancing parsimony with expressiveness.

### G.3. Simulation Integration

The discovered equations provide two pathways for improving the Koch's model without requiring the neural network at inference time.

**Direct Correction.** The most straightforward approach uses the discovered missing physics equation directly. The modified governing equation becomes:

$$\frac{\partial u}{\partial t} = [\text{Koch's terms}] + \gamma \cdot h(u, u_x), \tag{41}$$

where $h(u, u_x) = 0.72 - 3.79u + 6.24u^2 - 3.33u^3 + 0.40u_x - 0.40u \cdot u_x$ is the discovered correction functional and $\gamma$ is a tunable scale factor that accounts for normalization differences between training data and simulation variables.

**Hierarchical Correction.** Alternatively, the LF and HF corrections can be applied separately, enabling independent tuning of large-scale adjustments versus fine-scale dynamics:

$$\frac{\partial u}{\partial t} = [\text{Koch's terms}] + \gamma_{\text{LF}} \cdot f(\Delta u, u) + \gamma_{\text{HF}} \cdot g(u_{\text{HF}}, u_{\text{LF}}), \tag{42}$$

where $f$ and $g$ are the discovered LF correction and HF dynamics, respectively. This approach provides finer control but requires tracking both components during simulation.

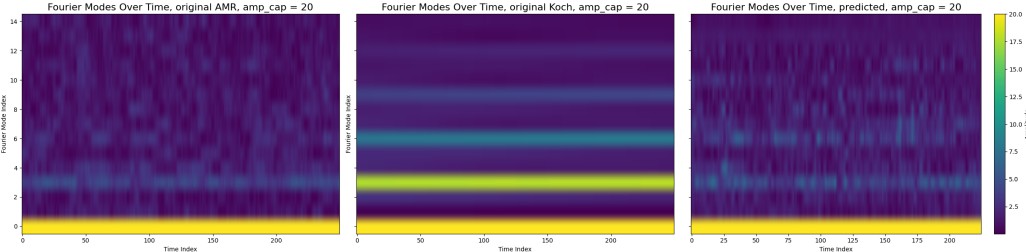

*(a)* Fourier modes for three co-rotating waves. Left: HF dataset; Middle: LF simulation; Right: Cheap2Rich reconstruction

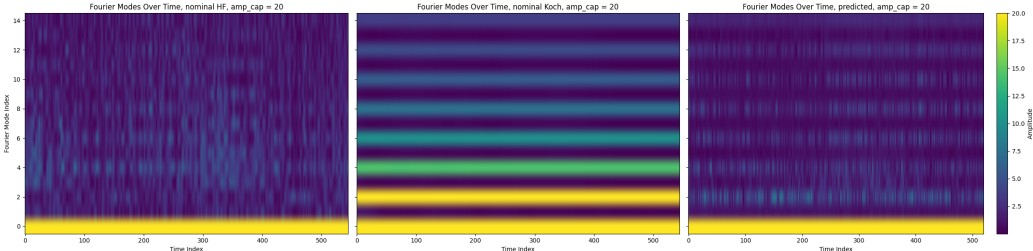

*(b)* Fourier modes for two co-rotating waves. Left: HF dataset; Middle: LF simulation; Right: Cheap2Rich reconstruction

*Figure 16.* First 15 Fourier modes and their development in time. The amplitudes are clipped at 20 for better visualization.

### G.4. Forward Integration Validation

To validate that the discovered equations constitute operable models, we forward-integrate the missing-physics equation (13) on top of Koch's 1D model. Sparse sensor measurements ($p = 25$) periodically provide partial observations of the discrepancy, serving as initial conditions for the correction PDE, which is then propagated forward between sensor updates using cosine-blended overlapping windows. Figure 15 presents the results. The SINDy-corrected Koch model achieves RMSE = 0.139, a 68% improvement over uncorrected Koch (RMSE = 0.434), while Cheap2Rich additionally captures fine-scale amplitudes (RMSE = 0.103, 76% improvement). The six-term polynomial PDE contains no neural network components yet bridges most of the fidelity gap, and the rollout remains numerically stable throughout the integration.

### G.5. Connection to DA-SHRED Framework

The SINDy-based discovery presented here extends the methodology of (Bao & Kutz, 2025) to the multi-scale setting. While the original DA-SHRED framework demonstrated discrepancy modeling for systems with a single learned correction, the multi-scale architecture provides a natural decomposition that yields richer physical insight:

- The **LF correction** captures what the latent-space alignment contributes—primarily amplitude and baseline adjustments that align the simulation manifold with reality.

- The **HF dynamics** reveal the structure of physics entirely absent from the simplified model, including the coupling between fine-scale fluctuations and the dominant wave structure.

- The **direct missing physics** provides a single equation summarizing all corrections needed, suitable for integration into simulations where the model discrepancy is perturbative—i.e., when the simulation is not far from reality.

This combined discovery approach—enabled by the Cheap2Rich architecture—provides both interpretability and actionable model corrections, advancing beyond pure reconstruction toward genuine physics discovery.

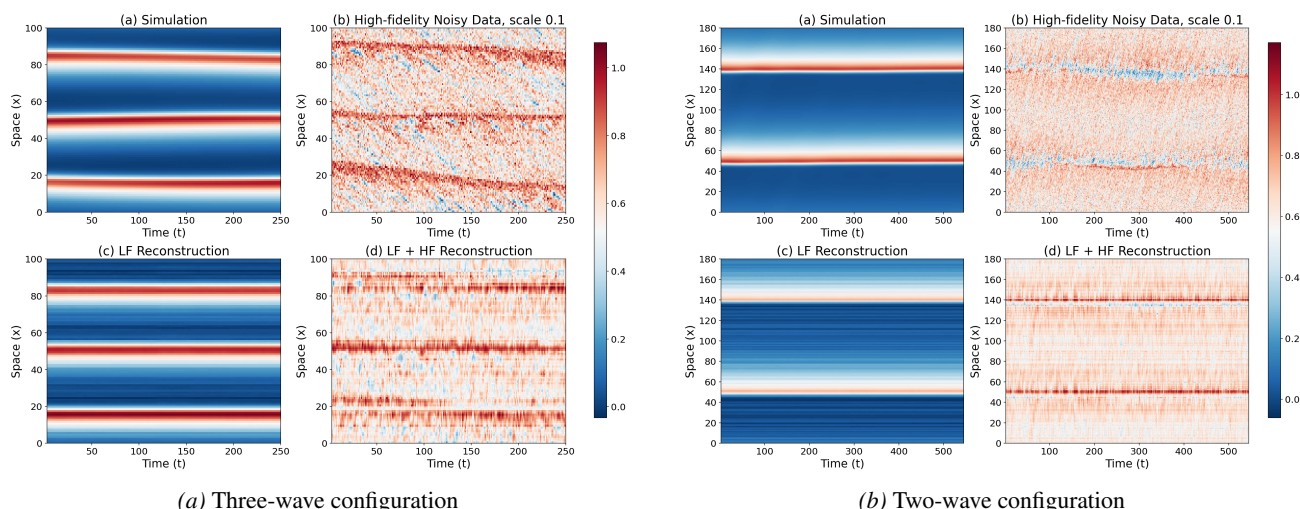

*(a)* Three-wave configuration                    *(b)* Two-wave configuration

*Figure 17.* Robustness test: Cheap2Rich deployed on noise-corrupted ($\sigma = 0.1$) high-fidelity sensor data. (a) Three-wave configuration, trained on 3-wave Koch prior. (b) Two-wave configuration, trained on 2-wave Koch prior. In both cases the HF pathway recovers the dominant wave structure from noisy sparse sensors. Two-wave metrics (mean $\pm$ std, 3 seeds): DA-SHRED only achieves RMSE $= 0.504 \pm 0.002$, SSIM $= 0.093 \pm 0.008$; Cheap2Rich achieves RMSE $= 0.125 \pm 0.006$, SSIM $= 0.241 \pm 0.019$ — a 75% RMSE reduction.

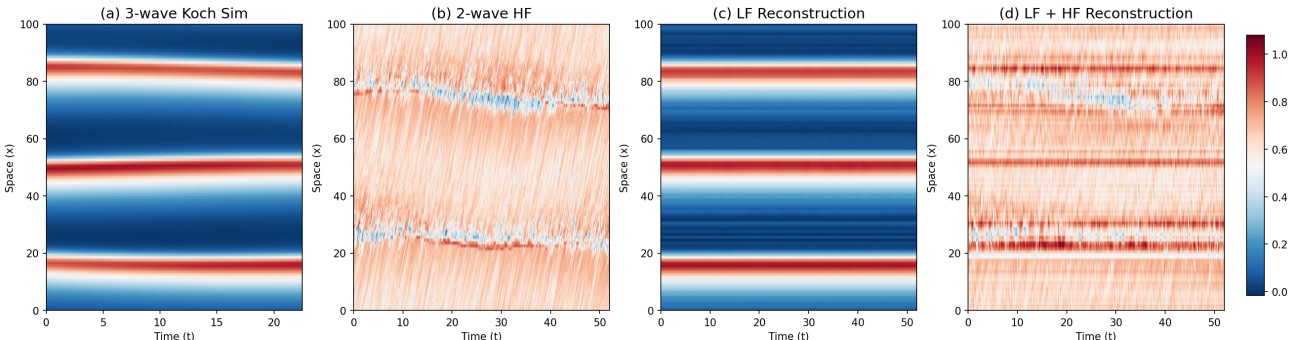

*Figure 18.* Cross-condition deployment: Cheap2Rich trained on a 3-wave Koch prior and deployed on 2-wave high-fidelity sensor data. (a) 3-wave Koch simulation (LF prior). (b) 2-wave high-fidelity ground truth. (c) LF reconstruction, which inherits the mismatched 3-wave topology. (d) Full Cheap2Rich LF+HF reconstruction. Despite the topological mismatch, the HF pathway achieves 76.9% RMSE reduction (0.458 vs. 0.106), though a residual third-wave artifact reduces SSIM (0.246 vs. 0.364 in the matched case).

## H. Spectral Comparison of Low- and High-Fidelity Models

Figure 16 compares the first 15 Fourier modes and their temporal evolution across three datasets: the high-fidelity simulation, the Koch LF model, and the Cheap2Rich reconstruction. Results are shown for both the three-wave (panel a) and two-wave (panel b) configurations.

The Koch model accurately captures the dominant modes but lacks the higher-harmonic content present in the high-fidelity data. The Cheap2Rich reconstruction recovers this missing spectral structure, confirming that the HF pathway learns the correct spectral corrections rather than introducing spurious energy.

## I. Cross-Condition Robustness

To evaluate generalization beyond the three-wave configuration used for primary evaluation, we train Cheap2Rich on a two-wave Koch prior ($s = 0.06$) and deploy it on two-wave high-fidelity data with $\sigma = 0.1$ sensor noise. This tests both cross-condition generalization and noise robustness simultaneously.

Figure 17 presents the reconstruction results. In the three-wave case (panel a), Cheap2Rich recovers the dominant wave

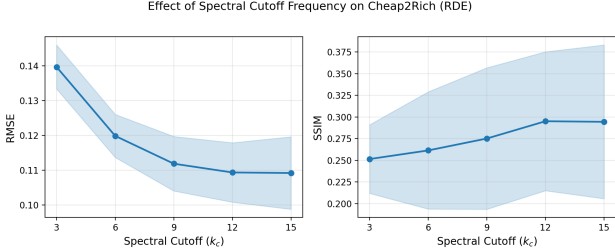
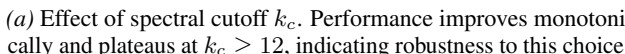

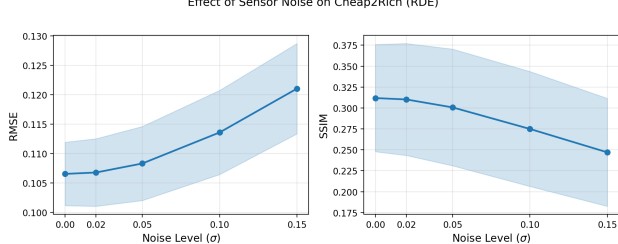

*(a)* Effect of spectral cutoff $k_c$. Performance improves monotonically and plateaus at $k_c \geq 12$, indicating robustness to this choice.

*(b)* Effect of sensor noise $\sigma$. Trained on clean data, deployed on noisy sensors. Performance degrades gracefully, consistent with the spectral cutoff providing noise protection.

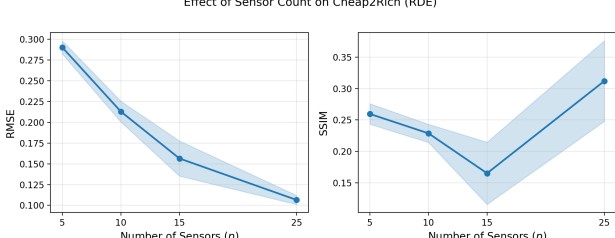

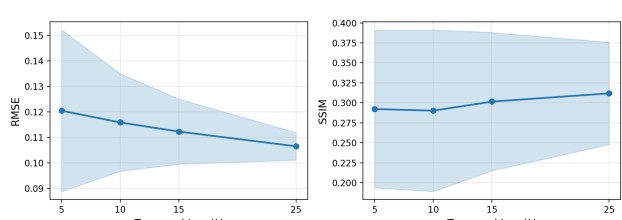

*(c)* Effect of sensor count $p$. RMSE improves steadily with more sensors.

*(d)* Effect of temporal lag $L$. Longer sensor histories yield slight improvement, with diminishing returns beyond $L = 15$.

*Figure 19.* Ablation studies on Cheap2Rich hyperparameters (3-wave co-rotating RDE). Each point reports the mean over 3 random seeds; shaded regions represent $\pm 1$ std.

structure despite sensor noise corruption. In the two-wave case (panel b), the framework achieves RMSE $= 0.125 \pm 0.006$ and SSIM $= 0.241 \pm 0.019$ (mean $\pm$ std over 3 runs), compared to DA-SHRED's RMSE $= 0.504 \pm 0.002$ and SSIM $= 0.093 \pm 0.008$—a 75% RMSE reduction. These results confirm that the multi-fidelity architecture generalizes across wave-mode configurations and is not overfit to the three-wave topology.

### I.1. Cross-Condition Deployment with Topological Mismatch

We further evaluate Cheap2Rich under a deliberately mismatched prior topology: the model is trained on a 3-wave Koch prior ($s = 0.07$) and deployed on 2-wave high-fidelity sensor measurements without retraining. This constitutes a worst-case scenario in which the LF model's wave count does not match the deployment reality.

Figure 18 presents the results. Despite the fundamental topological mismatch, the HF residual pathway compensates for the discrepancy using only sparse sensor measurements, achieving a 76.9% RMSE reduction ($0.458$ vs. $0.106$)—nearly identical to the matched-topology result. However, the reconstruction exhibits a residual artifact at the spatial frequency of the prior's third wave, resulting in lower SSIM ($0.246$ vs. $0.364$ in the matched case). This artifact arises because the additive LF+HF decomposition requires the HF pathway to simultaneously cancel the spurious third-wave structure and reconstruct the correct two-wave content—a harder task than pure fine-scale correction.

We emphasize that this mismatched scenario is not the intended operating regime: Koch's model generates any wave-count configuration in seconds by varying a single parameter (the mode number), so deploying with a matched prior is trivially inexpensive. The cross-condition result nevertheless demonstrates the robustness of the HF pathway and reveals a practical mechanism for mode-switch detection: when a topological mismatch exists, the HF pathway's spectral corrections and residual magnitudes differ observably from the matched-topology case, providing a diagnostic signal that could trigger re-generation of the prior with an updated wave count.

## J. Ablation Studies

We conduct ablation studies on the key hyperparameters of the Cheap2Rich framework using the 3-wave co-rotating RDE configuration. Each experiment reports the mean over 3 random seeds; shaded regions in Figure 19 represent $\pm 1$ standard

deviation.

The spectral cutoff $k_c$ is swept over $\{3, 6, 9, 12, 15\}$ (Figure 19a). Performance improves monotonically and plateaus at $k_c \geq 12$, confirming that $k_c$ is a robust hyperparameter. The cutoff defines a shared spectral search region: the LF output is spectrally smooth due to the spectral bias of neural networks (Rahaman et al., 2019), and the HF pathway discovers sparse corrections within the $[0, k_c]$ band; the out-of-band penalty suppresses noise absorption at higher frequencies.

Sensor noise robustness is evaluated by training on clean data and deploying with additive Gaussian noise $\sigma \in \{0, 0.02, 0.05, 0.10, 0.15\}$ (Figure 19b). Performance degrades gracefully, with RMSE increasing from $0.107$ to $0.114$ at $\sigma = 0.10$, consistent with the spectral cutoff providing noise protection.

The effect of sensor count $p \in \{5, 10, 15, 25\}$ is shown in Figure 19c: RMSE improves steadily with increased spatial coverage. Temporal lag $L \in \{5, 10, 15, 25\}$ (Figure 19d) yields diminishing returns beyond $L = 15$, consistent with Takens' embedding theory.

