# OpenReview forum: "Cheap2Rich: A Multi-Fidelity Framework for Data Assimilation and System Identification of Multiscale Physics - Rotating Detonation Engines"
_ICML.cc/2026/Conference — ICML 2026 regular_

### Official Review · Reviewer_XDzq · 2026-03-10

**Soundness:** 2
**Presentation:** 3
**Significance:** 2
**Originality:** 3
**Overall Recommendation:** 3
**Confidence:** 4

**Summary:**

A multi-fidelity framework called Cheap2Rich is introduced in this work. It reconstruct high-fidelity (HF) state spaces from sparse sensor data by bridging the gap between fast, low-fidelity (LF) priors and complex RDE dynamics. The framework implements two stage approach. In the first stage, a GAN-aligned SHRED model is used and in the second stage HF pathway that learns spectrally-sparse corrections from residuals. Additionally, SINDy is applied to the outputs to discover governing equations for the discrepancy terms to ensure the reconstructed dynamics remain physically interpretable. The advantage of the method is the integration of spectral constraints that help capture high-frequency physics that traditional 1D proxy models usually miss.

**Compliance With Llm Reviewing Policy:**

Affirmed.

**Final Justification:**

The paper presents a well-motivated multi-fidelity framework for reconstructing high-fidelity RDE dynamics using a structured LF/HF decomposition. The physical intuition and spectral separation are principled and meaningful contributions.

The rebuttal addresses several concerns, particularly by validating the SINDy-discovered equations via forward integration and clarifying RMSE inconsistencies. These additions strengthen the empirical support.

However, key issues remain. The comparison with the supervised MLP baseline does not clearly establish when the proposed method is preferable under limited HF supervision. The role of the GAN remains insufficiently justified, and no comparison with simpler alternatives is provided. Additionally, evaluation on a small, temporally correlated dataset limits confidence in generalization.

Overall, while the work has merit, these concerns reduce my confidence in its current form. I therefore maintain my weak reject recommendation.

**Key Questions For Authors:**

1. Appendix F shows a simple MLP can outperform the proposed model. Under what conditions especially as a function of available HF training data does Cheap2Rich become the preferable choice?
2. Since there is a minimal improvement from the GAN, what is the advantage does the adversarial objective contribute that a direct MMD or feature-matching loss would not?
3. Can the model transfer to different operating conditions (e.g., two-wave modes) without retraining? I think you should demonstrate the cross-conditional generalization of the developed framework.
4. How sensitive are the results to the cutoff frequency? Was this tuned on the validation set, and does it risk overfitting given the small sample size?
5. Have the SINDy-discovered equations been forward-integrated to confirm they capture stable discrepancy dynamics?

**Limitations:**

The authors identify the 1D scope as a future direction, but several technical gaps remain. The RMSE improvement is not clearly superior to simple supervised baselines, which is not fully addressed in the main text. The GAN component adds significant complexity for marginal benefit, raising concerns about reproducibility and stability. Furthermore, the SINDy equations remain unvalidated by forward simulation.

**Strengths And Weaknesses:**

Soundness:
The physical motivation used to decompose RDE dynamics into dominant front physics and fine-scale injector corrections is quite interesting. The slow-fast decomposition between Koch’s model and the HF dynamics is theoretically well described. Enforcing spectral separation is a principled choice, and the emergence of HF structures at specific harmonics k \in {3, 6, 9} suggests the framework connects the dynamical systems to physical reality. However, there are a few things which are not clear in the results:
1. Section 4.3 notes minimal improvement for the LF pathway even though adversarial latent alignment introduces significant training complexity. This suggests the GAN may not be as critical to the reconstruction as claimed. I think comparing this against simpler distribution alignment strategies (like MMD or Optimal Transport) would clarify if the adversarial loss is justified.
2. Table 1 shows that the RMSE is improved by 74.9%, but Appendix F shows a simple MLP trained on the HF data achieves RMSE of 0.0937, which outperforms Cheap2Rich that has a RMSE of 0.1031. This needs to be clarified. Also, the framework needs to be checked against this supervised baseline.
3. The discovered equations 11, 12, and 13 are presented as interpretable corrections but are never validated via forward integration.
4. The conclusion says RMSE has reduced by 80.9% but Table 1 reports 74.9% reduction. This needs to be resolved.

Presentation:
The paper is generally well-written and the figures are descriptive and informative. I liked Figure 7. The spectral analysis illustrates the dual-pathway separation quite well. However, some issues reduce clarity. The notation summary (Table 3) is essential for Section 3 but is buried in the appendix. Additionally, describing the four-stage training pipeline across different sections and appendices makes it difficult to follow. I think moving the supervised baseline from Appendix F in the main text would also provide a more transparent evaluation of the framework.

Significance:
The paper addresses a genuine and practically important problem. Using cheap proxy models to avoid expensive HF simulations is a conceptually meaningful contribution, and the computational cost argument (weeks vs. minutes) is interesting. The integration of SINDy as a downstream tool is a useful extension for physics discovery. However, I think the significance of the framework can be improved. First, it has a small dataset with only 250 snapshots. Additionally after 80/20 split you get 50 test snapshots that are temporally contiguous, which means the test set is almost certainly highly correlated with training data. Therefore, I think training Cheap2Rich on the three-wave setting and evaluating reconstruction quality on a two-wave or one-wave configuration would be helpful. Also, the Koch's model already supports one, two, and three-wave configurations (as shown in Figure 8) by varying the single parameter s.

Originality:
The combination of SHRED-based latent sensing, GAN alignment, and spectral decomposition into a unified pipeline presents a novel approach to RDE data assimilation.
The key insight is that enforcing spectral separation through complementary frequency constraints offers an elegant solution to the observability gap. That said, the individual components (SHRED, SINDy, GANs) are drawn from existing work, so the novelty lies primarily in their physically motivated combination rather than a fundamentally new technique.

---

> ### Author Rebuttal · Authors · 2026-03-31
>
> We thank the reviewer for the constructive feedback, and for recognizing the principled design of our model architecture. We address each concern below.
>
> - **W1-1 \& Q2: GAN vs. simpler alignment.**
>
> See *Review\#1, Q2* and *Review\#3, W1*
>
> - **W1-2: MLP baseline outperforms on RMSE.**
>
> See *Review\#3, W5*.
>
> To further contextualise, we also include FNO and DeepONet neural operator baselines under identical sparse sensing constraints (*Review\#2, W2*).
>
> - **W1-3 \& Q5: SINDy equations not validated via forward integration.**
>
> We have now conducted this validation. We forward-integrated the discovered missing-physics equation (Eq. 13) on top of Koch's 1D model. Sparse sensor measurements (25 points) periodically provide partial observations of the discrepancy, which serve as initial conditions for Eq. 13. The equation propagates the correction forward between sensor updates using cosine-blended overlapping windows, achieving RMSE $= 0.139$ - a 68\% improvement over uncorrected Koch (0.434) and bridging most of the gap to Cheap2Rich (0.103, 76\%). The rollout remains numerically stable with no divergence.
>
> Eq. 13 requires evaluation of $u$ and $u_x$ over the full spatial domain. In deployment, only sparse sensors are available, so a spatial reconstruction step is needed; we use linear interpolation from 25 sensors as a minimal baseline. In practice, the Cheap2Rich reconstruction itself could serve this role, creating a synergy: the neural network provides full-field estimates, and the discovered equation provides interpretable, equation-based propagation between updates. The two are complementary, not competing.
>
> The fact that a six-term polynomial PDE - with no neural network components, no learned latent spaces, and no access to the full high-fidelity state - can recover 68\% of the fidelity gap validates the system identification claim: the discovered equation is not merely an interpretive summary but an operable model that produces quantitative improvement when integrated forward.
>
> [*SINDy validation (LINK)*](https://tinyurl.com/ner3upv9)
>
>
> - **W1-4: RMSE inconsistency (80.9\% vs. 74.9\%).**
>
> Thank you for flagging this. 80.9\% refers to the full-dataset RMSE reduction; 74.9\% is validation-only. We will state both numbers clearly and consistently in revision.
>
> - **W2: Presentation.**
>
> We agree on first two points: Table 3 (notation summary) should appear closer to Section 3, the four-stage training pipeline should be consolidated. However, instead of presenting baselines with full-state supervision from Appendix F, we would present the added SOTA baselines under the same information constraints in the main text for transparency. We will restructure these in revision.
>
> - **W3 \& Q3: Significance and cross-condition evaluation.**
>
> *See Review\#2, W3*. And we acknowledge the limited dataset size (250 snapshots) as an inherent constraint of the high-fidelity simulation cost ($>$2 million CPU hours).
>
> - **W4: Originality.**
>
> We appreciate the reviewer recognizing that the spectral separation is an elegant solution to the observability gap. We agree the novelty lies in the physically motivated combination. As discussed in *Review\#3, W1*, this combination enables capabilities that the individual components do not provide in isolation - in particular, scale-separated SINDy discovery and forward-integrable correction equations.
>
> - **Q1: When does Cheap2Rich become preferable over the MLP?**
>
> See *Review\#3, W5*. The MLP requires full-state supervision on HF data in loss computation during training, which costs months of supercomputing time to generate per operating condition. Cheap2Rich requires only sparse sensor measurements from the HF system - a fundamentally cheaper information requirement. The MLP wins on pointwise RMSE when abundant HF training data is available; Cheap2Rich wins on structural fidelity (SSIM) and is the only viable option when full HF states are unavailable.
>
>
> - **Q4: Sensitivity to $k_c$.**
>
> *See Review\#3, W1* for the detailed explanation of what $k_c$ controls, and *Review\#2, W4* for the full ablation suite. In brief: $k_c$ defines the spectral search bandwidth (not a hard LF/HF partition), performance plateaus at $k_c \geq 12$, and the method is robust to this choice.
>
> We thank the reviewer again for all the great suggestions, and we respectfully hope the additional experiments and clarifications will be considered in the final assessment.

---

> > ### Author Rebuttal · Reviewer_XDzq · 2026-04-02
> >
> > The paper proposes a multi-fidelity framework for reconstructing high-fidelity RDE dynamics using a structured LF/HF decomposition, which is a meaningful and well-motivated direction.
> >
> > My main concerns were around (i) validation of the SINDy-discovered equations, (ii) fairness of comparison with supervised baselines, and (iii) limited evaluation across operating conditions. The rebuttal improves the paper in some aspects. In particular, the forward integration of the SINDy equations, the clarification of the RMSE reporting inconsistency, and the additional cross-condition and noise robustness experiments are useful additions and strengthen the empirical support.
> >
> > However, some core concerns remain. The difference in supervision between Cheap2Rich and the MLP baseline is now clearer, but a more systematic comparison showing when Cheap2Rich becomes preferable under limited HF supervision is still missing. Additionally, the role of the GAN remains unclear. It introduces additional complexity while contributing minimal improvement, and no comparison with simpler alignment strategies is provided. The evaluation is also still limited to a small and temporally correlated dataset, which weakens the generalization claims.
> >
> > Overall, while the rebuttal improves the paper, it does not fully resolve the key concerns that affect my confidence in the method and its evaluation. Therefore, I maintain my weak reject recommendation.

---

> > > ### Author Response · Authors · 2026-04-08
> > >
> > > We thank the reviewer for acknowledging that the SINDy forward integration, additional experiments, ablations and noise robustness studies strengthen our claims. We address the remaining concerns below.
> > >
> > > MLP comparison under limited HF supervision: we would like to clarify the fundamental asymmetry here. The MLP requires full-state HF fields during training, whereas Cheap2Rich observes only sparse sensor measurements from the HF system. These are not two points on the same supervision spectrum, but qualitatively different information regimes. Our sensor count ablation already demonstrates that Cheap2Rich maintains meaningful reconstruction performance even at p=10 sensors, a regime where full-state supervision is by definition unavailable. In the multi-fidelity setting motivating the paper - where obtaining full HF states can require months of supercomputing time per operating condition - Cheap2Rich is not competing with the MLP but addressing a setting to which the MLP is not applicable.
> > >
> > > We appreciate the suggestion for a systematic comparison under limited HF data availability. Under identical sensing constraints, we have already demonstrated with additional baseline comparisons that Cheap2Rich outperforms more modern neural operator baselines (*Review\#2, W2*).
> > >
> > >
> > > Role of the GAN: we agree the GAN contributes minimally to reconstruction quality - as explicitly reported in Table 1 and Section 4.3. We do not present it as a critical component: it is mentioned only eight times in the manuscript, each time in the context of acknowledging its limited effect. The primary contribution instead comes from the HF pathway with spectral sparsity regularization, which provides the largest RMSE improvement. We retain the GAN simply as an alignment step inherited from DA-SHRED and plan to include MMD and OT alternatives in the codebase for publication. We respectfully suggest that its marginal contribution should not weigh heavily in the overall assessment of the paper, given the architecture's effectiveness is clearly driven by the HF pathway.
> > >
> > > Dataset size and generalization: the limited HF data size constraint reflects the extreme computational cost of the high-fidelity AMR simulation ($>$ 2 million CPU hours). This data scarcity is precisely the challenge motivating our multi-fidelity approach: Cheap2Rich is designed for settings where HF data is prohibitively expensive to generate, and the framework's value lies in maximizing reconstruction quality from minimal HF access. Within this constraint, we have expanded the evaluation to test robustness across: (i) four hyperparameter dimensions through systematic ablations, (ii) sensor noise levels up to $\sigma=0.15$ with minor degradation, (iii) a matched two-wave operating condition with comparable performance, and (iv) a cross-condition deployment with topological mismatch (3-wave prior $\to$ 2-wave HF, linked in our response to *Review\#2*). While we agree that evaluation on a broader HF dataset would further strengthen the paper, we believe these additions have substantially broadened the supporting evidence beyond the original evaluation under limited operating condition.
> > >
> > > We thank the reviewer for the constructive engagement throughout this process, and respectfully hope that the clarifications and additional evidence may help resolve the remaining concerns. We will incorporate all the updates in the revision.

---

### Official Review · Reviewer_oX7x · 2026-03-13

**Soundness:** 2
**Presentation:** 2
**Significance:** 2
**Originality:** 2
**Overall Recommendation:** 3
**Confidence:** 3

**Summary:**

The paper studies state reconstruction for rotating detonation engines by combining a cheap 1D RDE surrogate (Koch’s model) with a learned correction pipeline that splits reconstruction into a low-frequency component and a high-frequency residual. The LF branch is essentially DA-SHRED-style latent adaptation with a GAN and explicit low-pass filtering; the HF branch predicts sensor-residual corrections with a sparsity-promoting spectral regularizer and a deformation module. The paper further applies SINDy to the learned discrepancy terms and argues that the recovered sparse dynamics are physically interpretable and useful for identifying missing injector-related physics. Empirically, the method is evaluated on one 1D projection of a high-fidelity 3D RDE simulation using sparse sensors, and the paper reports strong gains over a direct sim-to-real transfer baseline and improved SSIM over several appendix baselines.

**Compliance With Llm Reviewing Policy:**

Affirmed.

**Final Justification:**

The added ablations strengthen the empirical case, however, lack of novelty is issue for me.

**Key Questions For Authors:**

What exactly is the novelty relative to DA-SHRED, especially given that DA-SHRED already frames sim-to-real discrepancy modeling with SHRED, includes SINDy, and already reports an RDE example?

Can the authors clarify the GAN training objective in Eqs. (5)–(6)? As written, the equations and prose do not agree on whether the discriminator sees real-sensor latents or simulation latents on the 'real' side. If this is just a typo, please give the correct objective; if not, explain the actual training protocol.

Why should the HF branch be trusted as a full-field discrepancy model when it is trained only through sensor residual supervision? Please provide an ablation or argument showing that the full-field HF output is not just one of many unconstrained extrapolations consistent with the sensor residuals.

Can the authors evaluate on more than one operating condition / wave regime / sensor configuration, and with a stricter temporal split? Right now the evidence comes from one 250-snapshot sequence and one configuration. Showing robustness across multiple regimes or out-of-condition tests would materially strengthen the paper.

Can the authors validate the discovered SINDy equations beyond interpretive plausibility? For example, can the recovered equations be integrated forward, or inserted back into the cheap model, to show improved predictive fidelity?

**Limitations:**

No. The paper should more explicitly discuss that the target 'reality' is a high-fidelity simulation rather than experimental data, that the evaluation is confined to a single preprocessed 1D projection and one operating condition, that stronger appendix baselines reduce the headline advantage, and that the discovered SINDy equations are not yet validated as causal physical laws. On societal impact, the statement should engage more directly with dual-use implications for propulsion and defense rather than mentioning them only briefly.

**Strengths And Weaknesses:**

The main strength is that the paper tackles a genuinely important scientific-ML problem, i.e, reconciling a cheap physics proxy with richer multiscale dynamics in a regime where full-state observations are unavailable. The RDE setting is nontrivial, and the multi-scale intuition behind explicitly separating LF structure from HF discrepancy is sensible. I also think the qualitative reconstructions and the spectral view of the learned HF correction are the most compelling parts of the paper. The explicit cutoff/filtering and spectral penalty are concrete design choices rather than vague 'interpretability' language, and the paper does at least try to connect the learned HF content to physically plausible injector-modulated structure. The computational asymmetry between the expensive CFD source and the cheap surrogate also makes the application meaningful.

That said, I have serious soundness and originality reservations. On originality, the paper is much closer to an incremental variant of prior SHRED/DA-SHRED work than the presentation suggests. SHRED already exists as a sensing/reconstruction framework, and the very recent DA-SHRED paper already frames the problem as simulation-to-reality data assimilation with discrepancy modeling, explicitly embeds SINDy, and already includes an RDE example. Prior work in this literature also already studies data-driven modeling of rotating detonation waves. The genuinely new ingredient here seems to be the LF/HF decomposition with spectral regularization, not the broader DA + SHRED + discrepancy + SINDy recipe. The paper should position itself much more candidly against these immediate predecessors, especially DA-SHRED, because otherwise the novelty is overstated.

On soundness, the mathematical presentation has at least one nontrivial inconsistency. In Eqs. (5)–(6), the GAN objective samples both terms from 𝑝_sim, while the prose says the discriminator distinguishes simulation latents from transformed real-sensor latents and later says it classifies real sensor latents as real and transformed simulation latents as fake. Those are not the same training scheme. This may be a notation error, but in a paper whose LF adaptation depends on this step, such a mismatch matters. More broadly, the theory is mostly architectural description plus heuristic physical interpretation; there is no real argument that the LF/HF split is identifiable, stable, or even meaningfully separated under the chosen objectives.

The experimental design is the biggest weakness. The entire empirical case is built around one operating condition and one short sequence: 250 snapshots, 100 spatial points after aggressive dimensional reduction, 25 sensors, and an 80/20 split. The final evaluation appears to be on the last 50 snapshots of the same sequence, which is a weak generalization test for highly autocorrelated spatiotemporal data. There is no cross-condition evaluation, no robustness across sensor layouts or sensor counts, no noise study, no out-of-distribution test, no ablation on the cutoff 𝑘_𝑐, no ablation on the GAN, and no evidence that the SINDy equations improve forecasting or simulation when rolled back into the cheap model. As written, the work demonstrates interpolation on one preprocessed sequence, not robust sim-to-real closure.

A second major empirical issue is that the paper repeatedly talks about 'real sensors' and 'sim2real', but the target appears to be a high-fidelity simulation projected to a 1D ring, not actual experimental measurements. That is a valid multi-fidelity reconstruction problem, but it is not the same evidentiary bar as real-world sim-to-real assimilation. The claims should be narrowed accordingly.

The baseline story is also weaker than the main text suggests. Table 1 compares mainly against the raw sim-to-real gap and LF-only DA-SHRED, which makes the 74.9% gain look stronger than it is. In the appendix, an MLP trained on high-fidelity data only attains a better RMSE (0.0937) than Cheap2Rich (0.1031), and the paper itself concedes that Cheap2Rich’s aggregate RMSE is only “comparable to the baselines,” with its advantage being mainly SSIM/visual sharpness.

---

> ### Author Rebuttal · Authors · 2026-03-31
>
> We thank the reviewer for the detailed and engaged assessment. We have taken each concern seriously and responded with new experiments and clarifications. We respectfully disagree with the reviewer’s characterization that the paper should be positioned as an extension of DA-SHRED. The new results - particularly the neural operator baselines (*Review\#2, W2*), the ablation studies (*Review\#2, W4*), and the SINDy forward integration (*Review\#4, W1-3*) - show that the proposed multi-scale decomposition addresses a modeling challenge not resolved by DA-SHRED and other SOTA baselines, and therefore constitutes a substantive methodological contribution rather than incremental extension. We address each point below.
>
> - **W1 \& Q1: Novelty relative to DA-SHRED.**
>
> DA-SHRED learns a single perturbative correction in latent space. DA-SHRED's RDE example concerned a vanilla 1D transient stage of the cheap LF proxy, not the multi-fidelity reconstruction problem with a structurally different high fidelity dataset that we address here. In our three-wave experiment, the DA-SHRED output (Figure 6(c)) captures the correct front locations and propagation speeds but *cannot* reproduce any fine-scale injector-driven structure - it is spectrally smooth. This is not an artifact of our architecture: neural networks are known to exhibit spectral bias toward low-frequency structure (Rahaman et al., 2019), and the SHRED pipeline as well as the neural operator baselines all inherit this property.
>
> The cutoff $k_c$ is not a boundary that splits the spectrum into disjoint LF and HF bands. Rather, it defines a shared spectral region of interest. The LF output captures smooth broadband structure due to the spectral bias of the SHRED pipeline. The HF pathway searches within the same $[0,\,k_c]$ band for *sparse* corrections the smooth LF missed; the spectral sparsity regularizer (Eq. 28) enforces this parsimony. The penalty on energy above $k_c$ prevents the HF pathway from absorbing sensor noise - our noise robustness study (*Review\#2, W4*) confirms this: at 10\% sensor noise, RMSE increases only from 0.107 to 0.114. We acknowledge the phrasing beneath Eq. 28 ("sparse, low-frequency corrections") is misleading and will revise it.
>
> Our $k_c$ ablation (*Review\#2, W4*) shows monotonic improvement plateauing at $k_c \geq 12$, confirming $k_c$ is a practical rather than fragile hyperparameter. When physical priors are available, $k_c$ can be set tighter for noise protection; otherwise, a conservatively large $k_c$ does not degrade performance.
>
> - **W2 \& Q2: GAN objective.**
>
> We thank the reviewer for catching this - it is indeed a notation error. The first term of $\mathcal{L}_D$ should read $\mathbb{E}_{\mathbf{z} \sim p_{\text{real}}}[\log \mathcal{D}(\mathbf{z})]$, where $p_{\text{real}}$ denotes the distribution of latent codes obtained by encoding real-sensor histories through the frozen LF encoder. The implementation is correct; only the written equation contains the error. We will correct this in revision.
>
> - **W3 \& Q3 \& Q4: Experimental breadth and cross-condition evaluation.**
>
> We have conducted ablation studies covering $k_c$, sensor count, temporal lag, and noise robustness (*Review\#2, W4*). Additionally, we provide a two-wave configuration test with $\sigma = 0.1$ sensor noise (*Review\#2, W3*), achieving RMSE $= 0.125 \pm 0.006$ vs. DA-SHRED's $0.504 \pm 0.002$.
>
> - **W4: Sim2real terminology.**
>
> We acknowledge that "sim2real" in this context means low-fidelity to high-fidelity simulation, not simulation to physical experiment. The architecture is agnostic to the data source, but the current validation is multi-fidelity. We will narrow the terminology accordingly in revision.
>
> - **W5: Baseline comparisons and MLP.**
>
> The MLP baseline in Appendix F has full-state supervision on HF data in loss computation during training - a fundamentally different and easier problem setting. Cheap2Rich operates without access to HF full-state, using only sparse sensors from the HF system plus a cheap LF model. On structural fidelity, Cheap2Rich's SSIM also substantially outperforms the MLP.
>
> We further include FNO and DeepONet neural operator baselines under identical sparse sensing constraints (see *Review\#2, W2* for details, figures, and the updated baselines table). These results confirm that standard neural operators do not readily generalize to the multi-fidelity sparse-sensing setting.
>
> - **Q5: SINDy forward integration.**
>
> Addressed in detail in *Review\#4, W1-3*.
>
> We will add an explicit limitations section discussing: (i) the 1D scope; (ii) the dual-use implications for propulsion and defense; (iii) more challenging cross-condition generalizations (e.g. co-rotating $\to$ pulsing detonation) are yet to be explored. We thank the reviewer for the thoughtful evaluation of the additional materials and revisions, and respectfully hope that these clarifications may support a better overall assessment.

---

> > ### Author Rebuttal · Reviewer_oX7x · 2026-04-03
> >
> > Thank you for the detailed rebuttal. Several concerns are meaningfully addressed. In particular, the clarification that the GAN objective contains a notation error rather than an implementation error is helpful, and I appreciate the commitment to narrow the “sim2real” terminology to the multi-fidelity setting. The added ablations on noise,sensor settings,cutoff choice and the additional baselines and SINDy forward-integration results also strengthen the empirical case.
> >
> > That said, my concerns are only partially resolved. The main remaining issue is novelty/positioning relative to DA-SHRED. The manuscript itself states that it leverages SHRED and DA-SHRED, that the LF pathway follows standard DA-SHRED, and that the SINDy component extends the earlier DA-SHRED discrepancy-modeling methodology to the multi-scale setting. DA-SHRED already includes latent discrepancy modeling and SINDy-based discovery. So while I agree the explicit LF/HF decomposition may be useful and may improve performance, I am not yet convinced that the paper should be positioned as something other than a substantial extension of DA-SHRED unless this distinction is made much sharper in the revision.

---

> > > ### Author Response · Authors · 2026-04-08
> > >
> > > We thank the reviewer for acknowledging that the ablations, baselines, and SINDy validation strengthen our paper, and we respectfully suggest that the framing of Cheap2Rich as "a substantial extension of DA-SHRED" may understate the architectural, methodological, and capability-level differences it brings. The manuscript's wording was intended to acknowledge intellectual lineage, not to claim architectural equivalence. We will revise this language in the revision to make the distinction unambiguous. Concretely:
> > >
> > > The experiments presented in the manuscript, together with the additional experiments in the rebuttal, have already provided evidence that Cheap2Rich is not reducible to DA-SHRED. We further substantiate this with the cross-condition experiment linked in our response to *Review\#2*. When trained on a 3-wave Koch prior and deployed on 2-wave HF sensors, the HF pathway compensates for the topological mismatch using only sensor residuals, achieving 76.9\% RMSE reduction. By contrast, DA-SHRED's smooth latent perturbation cannot bridge this gap - it would produce a smoothly perturbed 3-wave field (as seen in the 3rd panel), since neural networks' spectral bias (Rahaman et al., 2019) confines latent corrections to the LF manifold. The HF pathway with spectral sparsity regularization is precisely the architectural ingredient that enables this capability.
> > >
> > >
> > > Similarly, the SINDy forward integration (Review 4, W1-3) highlights a practical consequence of this architectural difference. DA-SHRED's smooth discrepancy yields SINDy terms that capture only the low-frequency component of the missing physics - insufficient when the fidelity gap spans multiple spectral scales, as in the RDE setting. Cheap2Rich's explicit spectral decomposition provides SINDy with a richer, scale-separated discrepancy. The decomposition is what makes the discovered equations operable at scale, not just tractable in principle. We hope the reviewer will consider these distinctions in the final assessment.

---

### Official Review · Reviewer_XTjZ · 2026-03-13

**Soundness:** 2
**Presentation:** 2
**Significance:** 2
**Originality:** 2
**Overall Recommendation:** 3
**Confidence:** 3

**Summary:**

This paper presents "Cheap2Rich," a multi-fidelity data assimilation (DA) framework designed to bridge the "sim-to-real" gap in complex multiscale physical systems, specifically Rotating Detonation Engines (RDEs). The framework leverages a two-path architecture: 1) a low-frequency (LF) backbone trained on computationally inexpensive simulation priors and aligned to real sensor data via a latent GAN, and 2) a high-frequency (HF) pathway that learns fine-scale discrepancies from sensor residuals using spectral sparsity constraints. Finally, the authors apply Sparse Identification of Nonlinear Dynamics (SINDy) to the learned corrections to discover interpretable governing equations for the missing physics, providing a pathway for model improvement.

**Compliance With Llm Reviewing Policy:**

Affirmed.

**Final Justification:**

The added results on the cross-condition mismatch experiment and the new neural operator baselines are helpful, and they partially address my concerns. However, I still believe the paper needs a clearer justification of how the high-frequency pathway handles spurious structures and a more explicit discussion of its limitations regarding residual artifacts and adaptive design choices in the main text. I would like to keep my score.

**Key Questions For Authors:**

- Regarding Dimensionality Reduction: The article attempts to investigate a broad context of multiscale physics; however, does the 3D-to-1D projection filter out critical 3D instabilities that would normally hinder data assimilation? Does this projection make the task significantly easier than it would be in a raw 3D setting?
- Robustness to Mode Switching: RDEs often experience mode switching (changes in the number of waves). If the observed sensor data reflects a mode switch not present in the LF simulation, can the Cheap2Rich framework capture this topological change solely through the HF residual path?
- Sensor Noise performance: Real-world sensors are prone to noise. Do the reported RMSE improvements and SINDy-discovered equations remain stable when 5%-10% Gaussian noise is added to the sensor inputs?
- Closed-loop SINDy Validation: If the discovered missing-physics terms (Eq. 13) are added back into the low-fidelity numerical solver, does a pure numerical rollout of the modified solver match the high-fidelity AMR ground truth? This "closing of the loop" is essential to validate the "system identification" claim.

**Limitations:**

yes

**Strengths And Weaknesses:**

Strengths: The framework successfully combines the robustness of low-fidelity physical priors (the Koch model) with the flexibility of deep learning to capture unresolved dynamics.

Weaknesses:

- Limited Scope of Validation: The results are primarily based on a 1D ring projection of 3D Adaptive Mesh Refinement (AMR) data. While this captures the primary detonation front, it does not fully prove the framework’s efficacy in full 3D complex geometries where non-periodic instabilities or shear layer effects might dominate.
- Narrow Baseline Comparisons: The evaluation focuses on comparing the proposed method against the authors' previous SHRED framework. The paper lacks a comparison with industry-standard DA methods (e.g., Ensemble Kalman Filter, EnKF) or broader physics-aware surrogates (e.g., DeepONet) under the same sparse sampling constraints.
- Topological Dependency: The alignment performance appears highly dependent on the low-fidelity model capturing the base topology (e.g., wave number). If the LF simulation predicts three waves while the reality has two, it is unclear if the latent-GAN alignment would fail or introduce non-physical artifacts.
- Hyperparameter Sensitivity: The choice of the spectral cutoff frequency ($k_c$) and the adversarial stability of the GAN are critical to the results, yet the paper lacks a sensitivity or robustness analysis regarding these parameters.

---

> ### Author Rebuttal · Authors · 2026-03-31
>
> We thank the reviewer for the assessment and specific, actionable suggestions. We address each point below.
>
> - **W1: Limited scope of validation (1D projection).**
>
> We acknowledge that the 1D azimuthal projection captures the dominant detonation-front dynamics but omits radial and axial instabilities. This choice was deliberate: it provides a controlled setting to demonstrate the multi-fidelity methodology on the physically most important degree of freedom for RDE characterization. Extension to 2D/3D is discussed in Section 6 as future work and has been demonstrated in concurrent work on remote sensing applications. We will narrow the "sim2real" terminology to "multi-fidelity reconstruction" as suggested by Review\#3.
>
> - **W2: Narrow baseline comparisons.**
>
> We thank the reviewer for raising this important point. To further contextualise this comparison, we have added FNO (Li et al., 2020) and DeepONet (Lu et al., 2021) neural operator baselines that operate under the same information constraints as Cheap2Rich - trained on simulation, deployed on GT sensors without full-state supervision. We evaluate two FNO variants - a vanilla FNO trained on full simulation fields and a masked FNO trained on sparse zero-filled inputs with an explicit binary mask channel - as well as a DeepONet with a branch net receiving the same time-delay sensor history as our LSTM unit.
>
> The results reveal a clear structural inadequacy of standard neural operators for this problem setting. FNO (vanilla), which has never seen sparse inputs during training, produces spectrally corrupted outputs on high fidelity sparse fields. FNO (masked) achieves SSIM = 0.258, confirming that FNO can learn reconstruction from sparse measurements, but only within the simulation distribution. On high-fidelity sensor inputs, DeepONet reproduces simulation-like spatial structures despite the HF sensor values it receives.
>
> [*Baseline comparison table (LINK)*](https://tinyurl.com/3vzuwrz2)
>
> [*Baseline comparison figure (LINK)*](https://tinyurl.com/ysdh4xsn)
>
>
> We would like to emphasise that Cheap2Rich's advantage arises from the multi-fidelity architecture itself by addressing problems that neural operators, by design, do not tackle. Standard FNO and DeepONet are powerful tools but lack the mechanism to bridge the multi-fidelity discrepancy. Also see *Review\#1, Q5* for the EnKF discussion.
>
> - **W3: Topological dependency (wave number mismatch).**
>
> We have now conducted a cross-condition robustness test: Cheap2Rich is trained on a two-wave Koch prior and deployed on two-wave high-fidelity data with 10\% sensor noise corruption, demonstrating that the framework generalizes to a different operating condition. The model achieves RMSE $= 0.125 \pm 0.006$ and SSIM $= 0.241 \pm 0.019$ (3 seeds), comparable to the three-wave result (RMSE $= 0.104 \pm 0.001$). Combined with the three-wave noise robustness test, these results confirm that Cheap2Rich operates across different wave-mode configurations and under sensor noise.
>
> We acknowledge that characterizing limits on more extreme topological changes (e.g., co-rotating $\to$ pulsing detonation) could be an important direction for future work.
>
> [*Robustness test (LINK)*](https://tinyurl.com/3c8jpwt4)
>
> - **W4: Hyperparameter sensitivity.**
>
> We have conducted additional ablation studies to address this concern. We now provide: (i) a $k_c$ ablation across $\{3, 6, 9, 12, 15\}$ showing monotonic improvement that plateaus at $k_c \geq 12$ (*see Review\#3, W1 for detailed interpretation*); (ii) a sensor count ablation across $p \in \{5, 10, 15, 25\}$ demonstrating graceful degradation with diminishing spatial coverage; (iii) a noise robustness study where the model is trained on clean data and deployed with $\sigma \in \{0, 0.02, 0.05, 0.10, 0.15\}$ Gaussian sensor noise, showing stable performance; (iv) a temporal lag ablation across $L \in \{5, 10, 15, 25\}$ consistent with Takens' embedding theory.
>
> [*Ablation studies (LINK)*](https://tinyurl.com/3wx88zc3)
>
> - **Q1: Does 3D-to-1D projection make the task easier?**
>
> See W1. The 1D projection retains the dominant azimuthal dynamics, which is where the multi-scale structure (front vs. injector physics) is fully present. Even in 1D, the added baseline results (FNO, DeepONet, MLP, see W2) confirm the task is non-trivial.
>
> - **Q2: Robustness to mode switching.**
>
> See W3.
>
> - **Q3: Sensor noise performance.**
>
> See W4 (iii).
>
> - **Q4: Closed-loop SINDy validation.**
>
> See *Review\#4, W1-3*.
>
> We respectfully invite the reviewer to reconsider the assessment in light of these additions. We do not claim to address the full scope of RDEs; rather, this work focuses on a specific methodological component that we believe is necessary for progress in data-driven multi-fidelity settings.

---

> > ### Author Rebuttal · Reviewer_XTjZ · 2026-04-04
> >
> > Thank you to the authors for the detailed rebuttal, the inclusion of the new baselines (FNO, DeepONet), and the extensive ablation studies. I appreciate these clarifications, but I still have some remaining core concerns.
> >
> > It seems my question regarding topological dependency (mode switching) was misunderstood or sidestepped. I specifically asked what happens if there is a mismatch—e.g., if the LF simulation predicts three waves while the reality has two. The authors responded by training on a two-wave prior and deploying on two-wave high-fidelity data. While this shows the model can work on a different matched operating condition, it does not answer whether the latent-GAN and HF residual path can successfully correct a fundamental topological mismatch between the prior and reality, which is a very common issue in RDEs.

---

> > > ### Author Response · Authors · 2026-04-08
> > >
> > > We appreciate the reviewer's acknowledgment that our rebuttal and additional materials helped address the concerns, and we thank the reviewer for clarifying this important point on cross-condition deployment. We have conducted a cross-condition experiment in which the Cheap2Rich model is trained on a 3-wave Koch prior and deployed on 2-wave high-fidelity sensor measurements - a direct test of the topological mismatch scenario described by the reviewer.
> > >
> > > [*cross-condition experiment (LINK)*](https://tinyurl.com/2c5u8vcx)
> > >
> > >
> > > Despite the fundamental mismatch between the prior topology (3 co-rotating waves) and the HF deployment reality (2 co-rotating waves), Cheap2Rich still achieves a 76.9\% RMSE reduction (0.458 $\to$ 0.106), with RMSE nearly identical to the matched-topology result. The HF residual pathway successfully compensates for the topological discrepancy using only sparse sensor measurements, without access to the 2-wave full state during training. We note that the reconstruction exhibits a residual artifact at the spatial frequency of the prior's third wave, resulting in lower SSIM (0.246 vs. 0.364 in the matched case). This artifact arises because the additive LF+HF combination requires the HF pathway to simultaneously cancel the spurious third-wave structure and reconstruct the correct two-wave content - a harder task than pure fine-scale correction.
> > >
> > > Also, we wish to emphasize that this worst-case scenario - deploying with a fundamentally mismatched prior - is not the intended operating regime of the framework. Koch's 1D RDE model is a cheaper model that runs in seconds on a laptop for any wave count by varying a single parameter (the mode number). When the deployment wave regime is known or detected, generating a matched n-wave prior is trivially inexpensive - the computational bottleneck is always at the high-fidelity dataset (weeks of compute), never the low-fidelity simulation. We have verified that deploying with a matched 2-wave Koch prior on the same 2-wave HF data produces clean reconstruction without the third-wave artifact.
> > >
> > > [*Robustness test (LINK)*](https://tinyurl.com/3c8jpwt4)
> > >
> > > Furthermore, the cross-condition experiment reveals a practical mechanism for mode-switch detection: when a topological mismatch exists, the HF pathway's spectral corrections and residual magnitudes differ observably from the matched-topology case, providing a diagnostic signal that could trigger re-generation of the prior with an updated wave count. This positions Cheap2Rich not as a static pipeline but as an adaptive framework in which the inexpensive prior can be refreshed as operating conditions evolve - a natural capability for real-time RDE monitoring where mode switching is a known operational concern.
> > >
> > > We thank the reviewer again for the constructive discussions, and we respectfully hope that the additional materials could help resolving the remaining concerns.

---

### Official Review · Reviewer_eXmZ · 2026-03-19

**Soundness:** 3
**Presentation:** 2
**Significance:** 3
**Originality:** 1
**Overall Recommendation:** 4
**Confidence:** 4

**Summary:**

This works proposed an extension of the SHRED framework. The probed framework assumes shared low frequency components between low fidelity and high fidelity model and learns the low frequency correction by Cheap2Rich data assimilation framework by available sparse sensor data combining with low-fidelity prior. The results proves the feasibility of a general multi-fidelity DA framework for PDE based systems.

**Compliance With Llm Reviewing Policy:**

Affirmed.

**Final Justification:**

My concern is mostly addressed by the additional comprehensive additional experiments. So I would like to adjust my score to be 4.

**Key Questions For Authors:**

1. Presentation improvement. The figure 5 is a detailed explanation of figure 2 about how the multifidelity combination works, but they are too far away with each other. Could you combine them together?

2. Why do you use a latent GAN block in the multifidelity fusion block? Do you have any numerical evidence to justify this choice?

3. The high level idea of separating dynamics by different frequencies is similar to Fourier Neural Operator or Wavelet Neural Operator. What would be the benefit compared to these existing work? There isn't any comparison with SOTA baseline so hard to justify the effectiveness.

4. "On a three-wave co-rotating configuration (See Fig. 2)" Fig 2 is architecture. Do you mean Figure 1?

5. Any comparison with classical DA methods(ENKS, ENKF)?

6. Could you include frequencies metrics in ad, e.g., plot for error spreading with frequencies. It would be better to evaluate if the model learn the high frequency correction reasonably well.

**Limitations:**

The author is encouraged to discuss the limitations in detail.

**Strengths And Weaknesses:**

Strength:

1.Significance: proposed a good use case (RDE) why multifidelity modeling is necessary.

2.Soundness: the submission is technically with empirical results supporting claims.


Weakness:

Soundness: However, there is one assumption not verified. The framework assumes the model correctly learns the high frequency correction while the low frequency feature is modeled by Koch model. It would be intuitive to see the frequency plot of both Koch model and full order model to verify that the models have same low frequency feature.

Presentation: Many details are omitted in main script. At least the components mentioned in Fig 5 should be explained in more details. For example, how is the "deform" done?

Novelty: Novelty is not obvious compared with existing work. It looks like an extended application for SHRED and SINDy framework.

---

> ### Author Rebuttal · Authors · 2026-03-31
>
> We thank the reviewer for recognizing the significance of the RDE use case and the technical soundness of the empirical results. In light of the concerns raised, we have added comprehensive new experiments - neural operator baselines, ablation studies, a two-wave robustness test, and SINDy forward integration - that we believe collectively demonstrate both the difficulty of the problem for existing methods and the principled design of the Cheap2Rich architecture. We address each point below.
>
> - **W1: Frequency plot of Koch vs. HF model.**
>
>
> We agree this is a useful diagnostic and have included it in revision for both 3-wave and 2-wave settings: [*spectral analysis (LINK)*](https://tinyurl.com/m4mesdsj)
>
>
> - **W2: Deformation details omitted from main text.**
>
> The "Deform" module applies a learned spatially-varying shift and amplitude modulation to correct wave propagation velocity mismatches (Appendix D.3, Eqs. 20-23). Specifically, a shift network predicts per-gridpoint displacements $\delta(x)$ bounded by $\pm 10$ grid points, and an amplitude network predicts a positive modulation field $a(x)$. The final HF output is obtained by warping the base spatial pattern with periodic boundary conditions and bilinear interpolation. We will move a concise summary into the main text in revision.
>
> - **W3: Novelty relative to SHRED/SINDy.**
>
> We respectfully note that the contribution of Cheap2Rich is the multi-fidelity LF/HF pipeline with spectral regularization - a new architectural principle that enables capabilities absent from any individual component. DA-SHRED learns a single perturbative correction and we have demonstrated its ineffectiveness in our multi-fidelity settings. (*More details see Review\#3, W1*) Cheap2Rich enforces a structured decomposition that yields physically distinct components (front physics vs. injector physics) and enables discovering SINDy models for the discrepancy. The SINDy forward integration results (*see Review\#4, W1-3*) further validate that this decomposition produces tractable equations instead of just interpretive summaries. That Cheap2Rich demonstrably outperforms neural operators under identical sensing constraints (see Q3) constitutes, in our view, a contribution beyond incremental extension.
>
>
> - **Q1: Figures 2 and 5 placement.**
>
> We agree and will consolidate them in revision.
>
> - **Q2: Why GAN - numerical justification.**
>
> The GAN provides minimal LF improvement - this is stated in Section 4.3 and Table 1. We do not claim the LF pathway as critical; the dominant contribution comes from the HF pathway and spectral decomposition. The real pivot in Cheap2Rich is the additional architecture needed for the multi-fidelity task - the DA-SHRED result is spectrally smooth by construction due to the spectral bias of neural networks (Rahaman et al., 2019). As demonstrated in Q3, the LF pathway could be easily substituted with existing deep learning architectures - all of them simply cannot resolve fine-scale structure regardless of the alignment strategy. We retain the GAN as a lightweight alignment step inherited from DA-SHRED and plan to implement MMD and OT alternatives in the codebase for publication, allowing users to select the most appropriate aligner for their setting. See *Review\#3, W1* for detailed discussion on our architectural choices.
>
>
> - **Q3: SOTA baselines.**
>
> See *Review\#2, W2*.
>
>
> - **Q4: Figure reference error.**
>
> Thank you - "See Fig. 2" should read "See Fig. 1." We will correct this in revision.
>
> - **Q5: Classical DA methods.**
>
> EnKF requires a forward model that can be ensemble-propagated; the point of Cheap2Rich is that the cheap Koch model, while effective at capturing dominant front dynamics, is known to exhibit structured model-form error due to unresolved injector physics. EnKF would propagate these model-form errors forward without a mechanism to learn structured corrections from the data.
>
> - **Q6: Frequency-resolved error metrics.**
>
> The $k_c$ ablation (*see Review\#2, W4*) serves as a frequency-resolved error analysis: it shows how reconstruction quality improves as the spectral search window widens, with the plateau at $k_c \geq 12$ indicating that corrections beyond contribute negligibly.
>
> - **Originality.**
>
> We would like to respectfully invite the reviewer to reconsider the originality score in light of the additional baselines and ablations. Existing methods do not readily generalise to the multi-fidelity sparse-sensing setting addressed here, and the ablation studies confirm that our architectural choices have a robust and systematic effect on reconstruction quality. The SINDy forward integration (*Review\#4, W1-3*) further demonstrates Cheap2Rich as an operable model - a capability not addressed by the baselines considered here.

---

> > ### Author Rebuttal · Reviewer_eXmZ · 2026-04-07
> >
> > Thank you for the rebuttal. Most of my concern is resolved and I would like to adjust my score.

---

> > > ### Author Response · Authors · 2026-04-08
> > >
> > > We sincerely thank the reviewer for the careful re-evaluation and are glad the additional experiments addressed the concerns. We will incorporate the updates in the revision.

---

### Decision · Program_Chairs · 2026-04-30

**Decision:**

Accept (regular)

**Comment:**

This paper proposes a multiscale data assimilation framework, Cheap2Rich, which employs two pathways operating at low and high frequencies, and uses SINDy to learn the bias term in an interpretable manner. The authors ultimately apply the method to a rotating detonation engine system and successfully reconstruct high-fidelity physical states.

The reviewers consider the problem addressed by the algorithm to be important, and find the proposed method to be sound. In particular, the underlying Koch model reliable, and it is well described. Overall, the paper is clearly written and the figures are readable. The learned qualitative reconstructions, as well as the learned high-frequency corrections, are also considered interesting.

At the same time, the reviewers also raised several concerns. These mainly include that the distinction between the proposed framework and SHRED and SINDy is not sufficiently clear, which casts doubt on its novelty. Because the data consist of one-dimensional annular projections, it is difficult to demonstrate the method’s effectiveness on higher-dimensional and more complex states. In addition, the dataset is not sufficiently diverse, making it hard to fully validate the method. The algorithm's performance appears to depend on the underlying topology, and the effectiveness of introducing a GAN as part of the framework was also questioned.

However, the authors' rebuttal addressed most of the reviewers' concerns, and overall I would lean toward a weak accept.